# Stratospheric aerosol injection can weaken the carbon dioxide greenhouse effect
Haozhe He [1] ✉, Brian J. Soden[2], Gabriel A. Vecchi [1,3] & Wenchang Yang [3]

Stratospheric aerosol injection is a proposed method for offsetting greenhouse gas-induced warming by introducing scattering aerosols into the lower stratosphere to reflect sunlight. Here we explore a potentially more efficient alternative: weakening the Earth's greenhouse effect by deploying absorptive aerosols in the upper stratosphere (~10 hPa). These aerosols warm the carbon dioxide emission level—where outgoing longwave radiation is most sensitive to temperature—thereby enhancing top-of-atmosphere infrared emission without altering atmospheric carbon dioxide concentrations. Idealized climate model simulations indicate that this approach can reduce global temperatures an order of magnitude more efficiently per unit aerosol mass than conventional scattering-based interventions. Although based on simplified model experiments lacking interactive aerosol processes and operational constraints, our results identify a distinct physical mechanism for climate intervention, arguing for further research into the impacts—especially potential unintended side effects—of injecting absorptive aerosols into the upper stratosphere as an alternative solar radiation management strategy.

Keeping global warming below the 1.5 °C threshold set by the Paris Agreement[1] requires dramatic decreases in the emissions of greenhouse gases (GHGs). The potential to rapidly reduce GHG emissions is real, but progress is slow and global GHG emissions continue unabated[2]. Considering these concerns, exploring alternative intervention strategies, including solar radiation management, is essential. One possible approach, stratospheric aerosol injection (SAI), involves artificially enhancing the Earth's planetary albedo by increasing the number of small reflective particles in the stratosphere and has been proposed as an effective means of reducing surface temperature increases on a global scale, supported by substantial modeling and empirical evidence[3–5]. Sulfur dioxide ($SO_2$) has been featured prominently in SAI research, with volcanic eruptions serving as a natural analog[6]. During such eruptions, millions of metric tons of $SO_2$ are injected into the stratosphere, where they oxidize to form sulfate ($SO_4$) aerosols, which cool the Earth—referred to as conventional $SO_4$ SAI in the following. Although the intervention could reduce surface temperatures and potentially ameliorate some risks posed by climate change, it does not address the underlying driver of climate change (i.e., increasing GHG concentrations in the atmosphere) and cannot compensate for the radiative effects of the increased atmospheric GHGs in a direct way.

In contrast to the conventional $SO_4$ SAI, we hypothesize that alternative approaches that involve weakening the atmospheric greenhouse effect by warming the upper stratosphere (~10 hPa) via the injection of absorptive aerosols [e.g., black carbon (BC)] may warrant consideration and address some of the potential shortcomings of traditional SAI. This strategy is motivated by theoretical and modeling evidence that the carbon dioxide ($CO_2$) greenhouse effect depends strongly on the upper stratospheric temperature[7,8], where the $CO_2$ absorption bands achieve unit optical depth, allowing for unimpeded emission to space and therefore making outgoing longwave radiation most sensitive to temperature changes at this level. These studies show that increasing the emission temperature at these levels enhances the net longwave emission of radiation to space, weakening the greenhouse effect and countering the $CO_2$-induced cooling within the stratosphere, suggesting any change in atmospheric composition that perturbs the upper stratospheric temperature could also impact the climate by modulating the $CO_2$ forcing at the top-of-atmosphere (TOA), even without changing the $CO_2$ amount. This approach of warming the upper stratosphere with BC injection differs fundamentally from most SAI approaches, which use reflective aerosols to cool the planet. While analytic and radiative transfer calculations demonstrate the feasibility of weakening the $CO_2$ greenhouse effect[7,8], such an approach has heretofore never been proposed or tested in coupled ocean-atmosphere or earth system models, and its potential relative to other traditional SAI approaches remains largely unknown.

[1]High Meadows Environmental Institute, Princeton University, Princeton, NJ, USA. [2]Rosenstiel School of Marine, Atmospheric and Earth Science, University of Miami, Miami, FL, USA. [3]Department of Geosciences, Princeton University, Princeton, NJ, USA. ✉e-mail: haozhe.he@princeton.edu

Admittedly, a few previous studies of SAI using BC aerosols have been performed[9–11]; however, these studies did not consider the above-mentioned impact of BC in modifying the greenhouse effect, by altering the thermal structure of the upper stratosphere, focusing instead on its ability to cool the planet by reducing the solar radiation at the surface, either through radiative forcing at the tropopause or surface level. However, effective radiative forcing (ERF) at the TOA provides a more comprehensive representation of the overall climate impact of a forcing agent, as it accounts for all atmospheric adjustments[12–17]. Moreover, these studies only considered the injection of BC aerosols in the lower stratosphere, where its impact on the longwave emission to space by $CO_2$ is small and where it has the deleterious effect of warming tropopause temperatures and enhancing input of water vapor to the stratosphere. Although some studies investigated the altitude dependence of BC and $SO_4$ aerosols[18–20], these studies primarily examined the radiative effects of aerosols in the troposphere.

This study evaluates the altitude dependence of stratospheric aerosol deployments with both reflective and absorptive aerosols, as well as the difference between their effectiveness in mediating future warming. Utilizing a series of global climate model simulations, we show that by placing BC aerosols in the upper stratosphere (~10 hPa), near the emission level of $CO_2$, one can optimize its impact on enhancing the longwave emission to space and minimize the potential negative effects of tropopause warming. We further demonstrate that this approach provides a more efficient radiative forcing per unit aerosol burden relative to conventional scattering-based SAI approaches, with corresponding increases in cooling efficiency. These findings, while based on idealized model experiments that omit aerosol microphysics and real-world constraints, highlight the need for further research into the feasibility, effectiveness, and potential risks of upper-stratosphere deployments using absorptive aerosols.

## Results

The atmospheric greenhouse effect fundamentally depends on two quantities: the concentration of GHGs and the atmospheric temperature structure. An analytical model demonstrates that the radiative heat trapped by $CO_2$ represents a swap of tropospheric emission for stratospheric emission and, therefore, can be understood in terms of a dependence on the emission temperature of both the stratosphere and troposphere[8]. The former refers to the temperature of the upper stratosphere, where the $CO_2$ absorption bands achieve unit optical depth, allowing for unimpeded emission to space and therefore making outgoing longwave radiation most sensitive to temperature changes, while the latter depends on surface temperature and free-troposphere relative humidity. Recently, He et al.[7] used this model in conjunction with detailed radiative transfer calculations to demonstrate the dominant role that changes in upper stratospheric temperature play in determining the magnitude of the radiative forcing and greenhouse effect of $CO_2$.

Figure 1 shows the sensitivity of the outgoing longwave radiation to the temperature perturbations within the stratosphere ranging from 100 hPa to 1 hPa. The results are obtained from a set of offline radiative transfer calculations using monthly climatological temperature and humidity profiles of preindustrial simulations as well as the specified temperature perturbations at individual layers, by a broadband radiative transfer model (SOCRATES[21,22]).

These results demonstrate the importance of the vertical location of warming on the emission of outgoing longwave radiation. For the same magnitude of perturbation, a change in temperature at a higher altitude has a larger impact on the outgoing longwave radiation. Optimal sensitivity occurs when the temperature perturbation coincides with the level where $CO_2$ optical depth is ~1 (upper stratosphere), allowing for unimpeded emission to space. Air temperature increases within the upper stratosphere provide a more efficient longwave emission to space. Thus, one can reduce the strength of the $CO_2$ greenhouse effect by reducing the concentration of $CO_2$ or warming the emission level of $CO_2$. Note that the same temperature perturbation in a $CO_2$-free atmosphere would result in only one-tenth of the

outgoing longwave radiation change (Supplementary Fig. 1). Therefore, most of the changes in outgoing longwave radiation shown in Fig. 1 are due to the modulation of the $CO_2$ greenhouse effect.

## Altitude dependence of effective radiative forcing from prescribed absorptive and reflective aerosols

To test the sensitivity to the vertical level at which the aerosols are deployed, individual ERF calculations are done by separately placing the aerosols in each of the highest 7 model levels and maintaining fixed aerosol concentrations throughout the simulations (*Methods*; Supplementary Tables 1 and 2). It is important to note that these simulations are highly idealized and do not include interactive aerosol sedimentation, dynamics, chemistry, or real-world operational constraints. Figure 2 shows the global ERF for a horizontally uniform deployment of 0.5 Tg BC aerosols (left) and 0.5 and 5.0 Tg $SO_4$ aerosols (right) separately. For the $SO_4$ deployments, the ERF exhibits little sensitivity to the vertical location of the aerosol layer, with an ERF of ~0.12 W m$^{-2}$ for 0.5 Tg of $SO_4$ and ~1.3 W m$^{-2}$ for 5.0 Tg of $SO_4$, suggesting a roughly linear relationship between $SO_4$ aerosol loads and the corresponding ERF, as simulated in the models without considering aerosol microphysics.

In contrast to the $SO_4$, the ERF for BC aerosols exhibits a very strong sensitivity to the vertical level of deployment with higher altitude deployment resulting in substantially higher ERF values. Peak ERF values of ~2 W m$^{-2}$, which are an order of magnitude larger than that obtained for the equivalent mass of $SO_4$, are found at the two highest levels (~3.5 and ~7.4 hPa). As shown below, this larger value of forcing results primarily from the increased longwave emission to space induced by the warming of the stratosphere. These peak forcing levels coincide with the emission level of $CO_2$ (i.e., unit optical depth) where the outgoing longwave radiation is most sensitive to warming (Fig. 1). The ERF rapidly diminishes below this level, becoming effectively zero around 70.1 hPa. Nearly identical results are found with the GFDL-AM2.1 model (Supplementary Fig. 2). The weak values of ERF when BC is placed in the lower stratosphere are consistent with previous studies that found BC to be an ineffective option for solar radiation management[10,11]. However, these studies only considered injection in the lower stratosphere. In contrast, when BC aerosols are deployed in the upper stratosphere, they deliver a much more efficient radiative forcing per unit aerosol burden (~12 times stronger) relative to conventional $SO_4$-based SAI approaches with a peak ERF of −1.67 W m$^{-2}$ from 0.5 Tg BC vs. −0.12 W m$^{-2}$ from 0.5 Tg $SO_4$.

Next, we decompose the ERF into individual shortwave and longwave components (Supplementary Fig. 3). For $SO_4$ aerosol deployments, both the shortwave and longwave components exhibit little sensitivity to the vertical location of the aerosol layer (Supplementary Fig. 3b). For BC, which exhibits a very strong dependence of the ERF on altitude, most of this dependence (over two-thirds) results from the longwave component (Supplementary Fig. 3a), with the remainder from the shortwave which is discussed further below.

Considering that aerosol optical depth is a useful metric for assessing the negative shortwave radiative forcing, particularly below the aerosol layer (e.g., at the tropopause or surface in climate intervention simulations), especially in the presence of absorption aerosols, we diagnose the aerosol optical depth here. This is despite our focus on radiative forcing at the TOA, as it provides a more comprehensive representation of the overall climate impact of a forcing agent[12–17]. BC exhibits approximately three times the aerosol column extinction optical depth per unit mass compared to $SO_4$ (Supplementary Fig. 4), primarily due to its additional absorption, despite having slightly weaker scattering (Supplementary Fig. 5). Notably, the optical depth of both BC and $SO_4$ aerosols shows little sensitivity to the vertical placement of the aerosol layer. This greater optical depth per unit mass of BC aerosols, particularly the absorption component, counterbalanced by their pronounced longwave effect, results in BC producing roughly three times more radiative forcing per unit optical depth than $SO_4$ — with this forcing arising exclusively from longwave radiation, without contributing to shortwave cooling at the TOA.

---

To further examine the altitude dependence of BC forcing, we decompose the ERF into the instantaneous radiative forcing (IRF; direct effects) and rapid adjustments (indirect effects) (Supplementary Fig. 6; See more details in *Methods*). Note that we sum up the longwave IRF and stratospheric temperature adjustment as stratospheric adjusted longwave radiative forcing in Supplementary Fig. 6a to avoid discussing their altitude dependencies separately. This decomposition reveals a strong altitude dependence in the stratospheric adjusted longwave radiative forcing, ranging from $-4.7\,\mathrm{W\,m^{-2}}$ in the uppermost layers to $-2.2\,\mathrm{W\,m^{-2}}$ in the lowest layer (Supplementary Fig. 6a). The shortwave IRF is basically the same for all aerosol layers (Supplementary Fig. 6b).

Rapid adjustments from tropospheric clouds are also strongly dependent upon the level of aerosol deployment. The longwave cloud adjustment drops from $-0.25\,\mathrm{W\,m^{-2}}$ when BC aerosols are placed at ~3.5 hPa (uppermost level) to $-1.43\,\mathrm{W\,m^{-2}}$ at ~70.1 hPa (lowest level). Similarly, the shortwave cloud adjustment strengthens from $0.73\,\mathrm{W\,m^{-2}}$

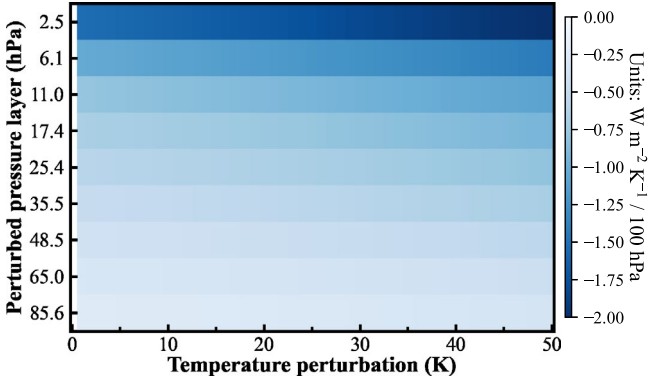

**Fig. 1 | Longwave radiative flux sensitivity at TOA to stratospheric temperature perturbations.** The net downward longwave radiative flux sensitivities at the TOA to per K temperature change ($\mathrm{W\,m^{-2}\,K^{-1}/100\,hPa}$) at individual pressure layers (hybrid-sigma level) obtained from a series of temperature perturbations ranging from 1 K to 50 K. The negative sensitivities to per K warming indicate more infrared radiation emitted out of the TOA by the stratospheric warming. Corresponding layer thicknesses normalize the sensitivity results and thereby represent the response to uniform perturbations in 100-hPa-thick layers.

when BC aerosols are placed at the uppermost layer to $1.23\,\mathrm{W\,m^{-2}}$ when the aerosols are placed at the lowest layer. This increasing cloud adjustment is responsible for the altitude dependence of the shortwave ERF (Supplementary Fig. 6). While the temperature dependence of the $CO_2$ infrared emission to space is based on fundamental physics and robust across models[7,8], cloud adjustments to anthropogenic forcing are poorly understood and differ considerably between models[16,17]. Thus, the cloud adjustments to stratospheric deployment of BC aerosols are also likely to be model-dependent.

**Effectiveness in mediating warming and related climate impacts**

To evaluate the effectiveness of these SAI deployments in mediating warming, we perform a series of coupled simulations (*Methods*) with both the high-resolution climate model (GFDL-CM2.5-FLOR, Supplementary Table 1) and the lower-resolution (more efficient) version model (GFDL-CM2.1, Supplementary Table 2). These climate intervention simulations are initialized from preindustrial conditions and conducted by prescribing 0.5 Tg BC and 5.0 Tg $SO_4$ aerosols at the second-highest level (~7.4 hPa), respectively. This level is considered the most effective deployment altitude due to its near-maximum ERF—only slightly lower than the highest value in GFDL-AM2.5 but the highest in GFDL-AM2.1. Additionally, deploying aerosols at this altitude is expected to be more practical, requiring less energy and logistical effort than reaching the highest level. The 0.5 Tg BC simulation induces slightly more cooling (~0.8 K) compared to the 5.0 Tg of $SO_4$ (~0.6 K) (Fig. 3a). The effectiveness in mediating warming is observed in both a single realization of the GFDL-CM2.5-FLOR (dotted line) and the 3-member ensemble mean of the GFDL-CM2.1 (solid line and shading), as well as in the 3-member ensemble means of intervention simulations initialized from both non-equilibrium and equilibrium $CO_2$ doubling cases using GFDL-CM2.1 (shown in Supplementary Figs. 7 and 8, though not discussed further to maintain a streamlined narrative). The difference in the cooling effects between that of 0.5 Tg BC and 5.0 Tg $SO_4$ aerosols is mainly attributed to their difference in ERF, as both sets of experiments have similar feedback values (Supplementary Fig. 9a).

The more substantial cooling in the 0.5 Tg BC simulations leads to a greater reduction in global rainfall (Fig. 3b), noting that both sets of forcing experiments exhibit similar rates of rainfall reduction per unit global temperature change (Supplementary Fig. 9b). Rapid precipitation decreases are found in both of these climate intervention simulations (Fig. 3b and

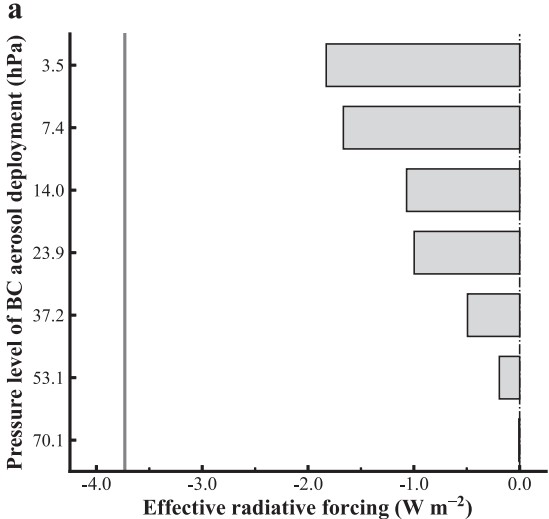
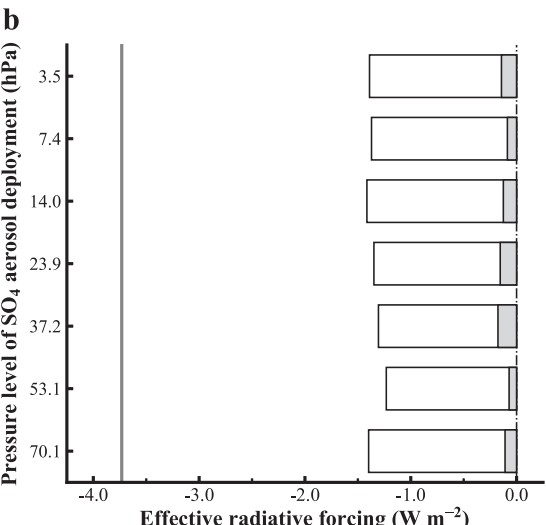

**Fig. 2 | Altitude dependence of effective radiative forcing from prescribed BC and $SO_4$ aerosols.** Effective radiative forcing (ERF) of horizontal-uniformly prescribed (**a**) 0.5 Tg BC aerosols and (**b**) 0.5 and 5.0 Tg $SO_4$ aerosols separately at each of the highest 7 sigma levels of aerosol inputs, respectively, simulated by the GFDL-AM2.5 model. These sigma levels are converted to approximate pressure levels by

multiplying by 1000 hPa and are shown in the figure. The bar represents the ERF calculated as the difference of net radiative fluxes at the TOA. The filled and open bars in (**b**) display the ERF results from 0.5 and 5.0 Tg $SO_4$ aerosols, respectively. The solid vertical lines represent the flipped ERF value of doubling $CO_2$.

Supplementary Fig. 9b). In particular, the fast precipitation decline in the BC simulations occurs without notable surface temperature changes. This contrasts with our initial expectation that upper stratospheric BC deployment would increase precipitation by counteracting the direct radiative effects of $CO_2$ — a departure from the precipitation decreases typically associated with tropospheric BC aerosols and their atmospheric absorption[23]. Typically, an increase in $CO_2$ suppresses fast precipitation by

reducing outgoing longwave radiation (or increasing atmospheric absorption) due to energetic constraints, alongside atmospheric stabilization[23,24]. Preliminary analysis suggests that this suppression may stem from anomalous infrared absorption in the lower troposphere, possibly driven by unexpected increases in water vapor.

Similar patterns of air temperature and precipitation change are also found between both intervention strategies. We focus on the last 30 years of these coupled simulations and plot the time-mean spatial patterns of both surface air temperature and precipitation changes per unit global temperature change (Fig. 4 and Supplementary Fig. 10). Both models exhibit delayed cooling over the Southern Ocean, resulting in an interhemispheric cooling contrast with more cooling over the Northern Hemisphere and less cooling over the Southern Hemisphere. The interhemispheric contrast is more noticeable in BC simulations using the GFDL-CM2.1 (Fig. 4), enhanced by the strong Arctic amplification and the more uniform cooling over the northern Pacific, which do not occur in the $SO_4$ simulations. Correspondingly, there is a general precipitation declining pattern, which is opposite to the "dry gets drier, and wet gets wetter" paradigm[25], and the effect of tropical ocean warming patterns[26] under $CO_2$ increase scenarios. There is pronounced drying over the tropics and wetting over the subtropics, with stronger wetting in the Southern Hemisphere accompanied by a southward shift of the Intertropical Convergence Zone. Consistent with the above-mentioned different cooling over the tropical Pacific, BC simulations show more uniform drying over the tropics, while $SO_4$ simulations display more drying over the eastern Pacific and tropical Atlantic, as well as an eye-catching wetting over the equatorial western Pacific.

Similar cooling patterns are also shown in the single realization of the GFDL-CM2.5-FLOR (Supplementary Fig. 10), especially for the BC simulation, except for the weaker "warming hole" over the Northern Atlantic in both simulations. In addition, noticeably inconsistent features are found over polar areas for the $SO_4$ simulation. Different from the ensemble-mean results of GFDL-CM2.1, GFDL-CM2.5-FLOR simulates a stronger Arctic amplification with comparable strength to that of BC simulation, but it is offset by even stronger delayed cooling over the Southern Ocean, resulting in a comparable global cooling rate. It is worth noting that identical tropical cooling patterns are found consistently across the two models, especially over the tropical Pacific, with a La Niña-like pattern for the BC simulations and missed cooling over the central Pacific for the $SO_4$ simulations, leading to identical precipitation responses. The consistent tropical responses probably suggest the role of radiative forcing in mediating warming via modulating the forced cooling patterns over the tropics, while the distinct responses over polar areas across the two models could probably be attributed to the internal variabilities, especially the low-frequency internal variability in models, considering the exact ocean model is adopted in the two climate models.

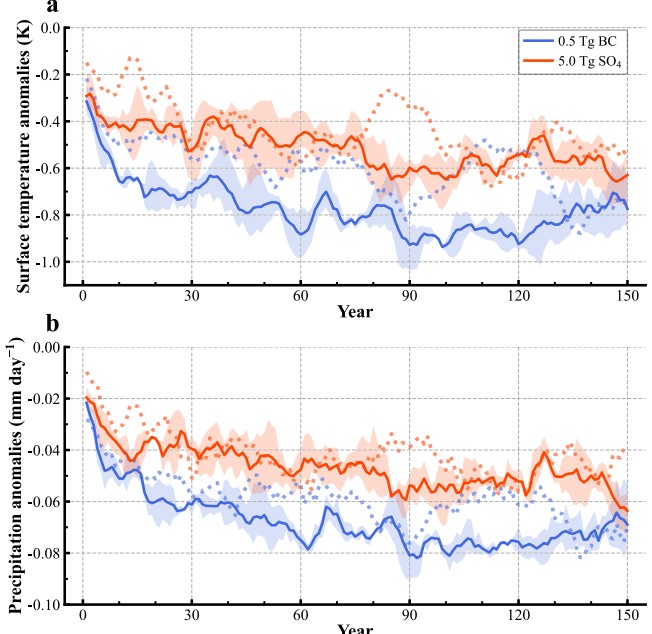

**Fig. 3 | Global mean surface air temperature and precipitation anomalies from climate intervention simulations.** Global mean (**a**) surface air temperature and (**b**) precipitation anomalies in climate intervention simulations forced with 0.5 Tg BC and 5 Tg $SO_4$ aerosols separately, by horizontal-uniformly prescribing the targeted aerosols at the second highest sigma level of aerosol inputs, beginning from the preindustrial control runs. Anomalies are taken relative to the climatological mean of the preindustrial control runs. The solid line represents the ensemble mean of 3 members from the GFDL-CM2.1 model with different initial conditions taken from the preindustrial control simulation. The corresponding light shading represents the range from minimum to maximum for every year. The dotted lines are the results of a single realization of the GFDL-CM2.5-FLOR model. A 5-year running mean is applied to each time series, in particular, to reduce the linear effects of El Niño-Southern Oscillation (ENSO) variability.

**Fig. 4 | Ensemble-mean spatial patterns of surface air temperature and precipitation change per unit global temperature change in response to aerosols.** The ensemble-mean time-mean spatial patterns of (**a**, **b**) surface air temperature (K K⁻¹) and (**c**, **d**) precipitation ($10^{-1}$ mm day⁻¹ K⁻¹) change per unit global temperature change, in response to the (**a**, **c**) 0.5 Tg BC and (**b**, **d**) 5 Tg $SO_4$ aerosols, horizontal-uniformly prescribed at the second highest sigma level of aerosol inputs, beginning from the pre-industrial control runs by the GFDL-CM2.1 model.

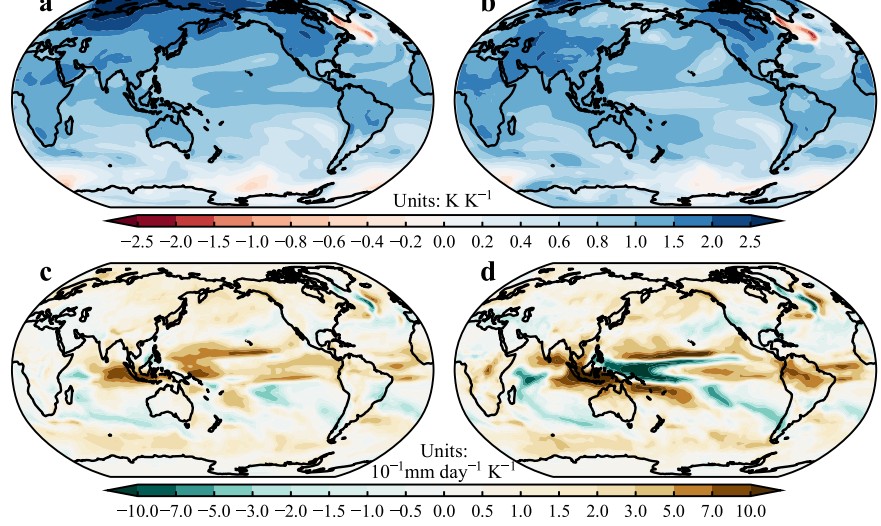

A concern of both intervention strategies is the warming of tropopause temperatures, which can trigger a cascade of negative consequences — including increased water vapor input into the stratosphere, reductions in stratospheric ozone concentrations[11,27], and disruptions to both stratospheric and tropospheric circulations[28–32]. As expected, the maximum atmospheric warming closely coincides with the location of the aerosol layer for both BC and $SO_4$, although BC aerosols generate substantially greater warming due to their higher absorptivity (Supplementary Fig. 11a). The higher and thinner the atmosphere, the greater the warming. It is important to note that higher temperatures may accelerate the catalytic reaction rates of ozone-depleting processes[33,34], potentially resulting in greater ozone loss, although these effects are not considered in our simulations, which prescribe ozone concentrations. Aerosols in the lower stratosphere induce more warming around tropopause and upper troposphere, weakening the "freeze-drying" process and consequently increasing the stratospheric water vapor mixing ratio[35,36] (Supplementary Fig. 11b). This water vapor increase contributes additional radiative forcing[37] and, more importantly, may catalyze ozone depletion through the formation of hydroxyl (OH) radicals[38]. The warming also leads to a larger increase in the saturation vapor pressure. Without substantially enhanced water vapor transport into the area, it is reasonable to expect a decrease in the relative humidity (Supplementary Fig. 11c). Similarly, cloud fraction decreases within the upper troposphere (Supplementary Fig. 11d). The lower we prescribe the BC aerosols, the more the cloud fraction decreases, likely due to the stability Iris hypothesis[39], which states that the increased stability reduces the magnitude of the radiatively driven clear-sky mass convergence at the height of anvil clouds, thus weakening convective detrainment at that height, leading to a reduction of the anvil coverage.

Beyond their direct radiative impacts, BC and $SO_4$ aerosols also influence stratospheric dynamics[28,29], warranting further consideration. Our preliminary results suggest that BC-induced warming in the upper stratosphere can modify both the Brewer-Dobson circulation and the polar vortex. Unlike sulfate aerosols, which accelerate the Brewer-Dobson circulation by enhancing tropical upwelling, BC-induced heating appears to slow it down, potentially altering stratospheric ozone transport and distribution (Supplementary Fig. 12). Seasonal variations in BC radiative heating may also strengthen the equator-to-pole temperature gradient in winter, reinforcing the polar vortex. This has implications for stratosphere-troposphere coupling, as a stronger polar vortex has been linked to a positive phase of the North Atlantic Oscillation, which in turn influences storm tracks, temperature anomalies, and precipitation patterns across mid-latitude regions, particularly over Europe and North America[40,41]. However, our model lacks interactive stratospheric chemistry and has a relatively low model top, limiting its ability to capture the full extent of these interactions, particularly given that the available monthly wind field outputs adopted to calculate residual circulation[42,43] smooth out short-term fluctuations and lose fine-scale temporal details. Nonetheless, our findings suggest that BC-induced upper stratospheric warming may influence large-scale circulation patterns in ways distinct from $SO_4$ aerosol geoengineering.

## Conclusion and discussion

Motivated by theoretical and modeling evidence that the $CO_2$ greenhouse effect depends strongly on the upper stratospheric temperature[7,8], we propose a potentially more effective approach to SAI that uses absorbing aerosols to induce a weakening of the Earth's greenhouse effect. The deployment of absorbing aerosols in the upper stratosphere increases the emission of infrared radiation out of the TOA. By warming the emission level of $CO_2$, where it reaches unit optical depth and therefore outgoing longwave radiation is most sensitive to its temperature changes, this process directly weakens the greenhouse effect by altering the thermal structure of the upper stratosphere rather than the concentration of GHGs.

Prescribing the same amount of aerosols separately at each of the highest 7 sigma levels of the aerosol input file and running the recommended ERF diagnostics, we found that the ERF of BC aerosols depends strongly on the altitude at which BC aerosols are positioned, while the ERF

of $SO_4$ aerosols exhibits little sensitivity to the vertical location of the aerosol layer. Here the stratospheric adjusted longwave radiative forcing initiated by BC aerosols, from the weakening $CO_2$ greenhouse effects and strengthening stratospheric temperature adjustment, dominates the altitude dependence of BC ERF.

By placing BC in the upper stratosphere near the emission level of $CO_2$, the BC aerosols deliver a more efficient radiative forcing per unit aerosol burden (~12 times stronger) relative to conventional $SO_4$-based SAI approaches. The models examined here have very similar climate sensitivities for both forms of SAI, thus BC aerosols also provide a more efficient cooling of the climate per unit aerosol mass. For the models examined here, GFDL-CM2.5-FLOR and GFDL-CM2.1, the reduction in global temperatures induced through the upper stratospheric BC aerosol deployment is more than an order of magnitude larger compared to the deployment of more traditional $SO_4$ aerosols. Note that these results stem from highly idealized model experiments that do not include interactive aerosol sedimentation, atmospheric dynamics, chemistry, or operational constraints. These simplifications limit the realism of the findings, especially with regard to aerosol transport, lifetime, and microphysical evolution. Therefore, while the results argue for further research into the possible benefits, costs, and unintended side effects of injecting absorptive aerosols in the upper stratosphere as a potential alternative strategy for solar radiation management, this line of inquiry should be pursued using more comprehensive modeling frameworks, guided by improved understanding of aerosol physical and chemical processes. Particular future emphasis should be on better understanding the optical, chemical, and photochemical interactions, along with the transport, lifetime, and size evolution of absorbing aerosols in the upper stratosphere, and for modeling studies with more comprehensive models that incorporate this improved knowledge. Further, these results indicate that the impact of volcanic and other natural stratospheric aerosols that may arrive to the upper stratosphere will be strongly dependent on the optical properties of these aerosols.

We note that neither of the atmospheric model configurations we consider (AM2.5 and AM2.1, and the respective coupled models) has interactive aerosols, stratospheric chemistry, or a well-resolved stratosphere (unlike, for example, GFDL's ESM4[44], or NCAR's WACCM6[45]). An advantage of our model configuration is that their efficiency and simple aerosol treatment allow us to explore the climate response to highly controlled aerosol changes. These experiments provide motivation for further, more computationally expensive, work (some ongoing) with more comprehensive models (e.g., models with interactive chemistry in the stratosphere, a better-resolved stratosphere, etc.) to better understand the full response of the system to absorbing aerosols in the stratosphere.

We acknowledge that key uncertainties remain in the transport, microphysical evolution, and sedimentation of absorbing aerosols. Our simulations prescribe aerosol concentrations uniformly at fixed altitudes, neglecting redistribution due to transport processes such as interactions with stratospheric circulation, as well as self-lofting and its seasonal variability driven by changes in incoming solar radiation. Since the Brewer-Dobson circulation plays a crucial role in aerosol transport, anomalous radiative heating could further shape aerosol distribution, modulating their lifetime and climatic impacts[46–48]. Additionally, aerosol microphysics—governing size and compositional changes through nucleation, condensation, coagulation, hygroscopic growth, and sedimentation—can considerably affect radiative properties and further influence residence time at the intended altitude[47,49,50]. As BC aerosols move downward, they may warm the lower stratosphere, reducing the cooling efficiency of the intervention. This warming could also alter temperature gradients, destabilizing the stratosphere and potentially affecting tropospheric circulation patterns. Moreover, the enhanced lower-stratospheric heating may increase water vapor concentrations, which could counteract the intended cooling effects by amplifying the greenhouse effect and exacerbating ozone depletion. In addition, when BC aerosols sediment onto snow and ice surfaces, they reduce surface albedo, causing the snow and ice to absorb more solar radiation rather than reflecting it. This effect accelerates melting, increases

surface temperatures, and triggers feedback mechanisms that could offset some of the cooling effects of BC aerosols.

On the other hand, BC aerosols also absorb ultraviolet radiation[51], which could locally reduce ultraviolet flux and mitigate ozone depletion by limiting ozone photodissociation, particularly for ozone below the aerosol layer. However, our model does not include interactive ozone chemistry, constraining our ability to fully quantify these effects. Future studies incorporating interactive aerosol dynamics, stratospheric chemistry, and higher model tops are essential for refining these findings and improving our understanding of BC-induced stratospheric changes.

Admittedly, delivering aerosols to the upper stratosphere presents major engineering challenges[52]. While feasibility assessments[53,54] suggest that reaching altitudes around 30 km remains difficult, continued technological advancements may render this more achievable. Current designs for specialized high-altitude aircraft indicate that deployment to around 20 km is already within reach, and further innovations could extend this range. Other options for delivering material to the stratosphere include rockets, ballistic payloads launched from the ground (e.g., artillery or rail guns), and balloons. In addition to direct delivery methods, the self-lofting effect of absorbing aerosols, particularly BC aerosols[55], offers a potential mechanism to sustain aerosols at higher altitudes. This effect relies on the aerosols absorbing solar radiation, heating the surrounding air, and generating upward motion, helping to maintain the particles aloft. However, the efficiency and duration of this process remain uncertain, particularly when factoring in seasonal variations in solar radiation and potential interactions with stratospheric dynamics[56]. Moreover, large-scale deployment raises critical concerns about scaling, cost, and logistics. Maintaining a steady, controlled aerosol layer would likely require continuous delivery, adaptive flight strategies, and rigorous monitoring to account for atmospheric variability and aerosol dispersion. These operational challenges, coupled with the need for specialized infrastructure and international coordination, underscore the importance of further research into both the physical science and engineering aspects of stratospheric aerosol delivery.

## Methods
### Models and experiments
To evaluate the effectiveness of this strategy, we perform a series of climate model simulations using atmospheric models with prescribed sea surface temperatures (SSTs) and sea ice concentration to explore the altitude dependence in stratospheric aerosol deployments with both reflective and absorptive aerosols. These fixed-SST simulations provide the recommended effective radiative forcing (ERF) diagnostics[13], including both the direct radiative effects of the SAI and the additional radiative effects from the climate response to the intervention, such as rapid adjustments, which are substantial for anthropogenic aerosols[16]. These fixed-SST simulations also help to understand how SAI deployment of absorbing aerosols modifies the temperature structure of the stratosphere and the net flow of radiation at the TOA.

A high-resolution global atmosphere model (GFDL-AM2.5 as configured and run in the Princeton University Tiger2 supercomputer[57,58]) is used for a comprehensive set of experiments (Supplementary Table 1). GFDL-AM2.5 is the atmospheric component of the global coupled climate models GFDL-CM2.5[59] and GFDL-CM2.5-FLOR[60]. The GFDL-AM2.5 has a finite volume cubed-sphere dynamical core at a global 50-km resolution ($180 \times 180$ grid points on each cube face) at 32 vertical levels[61].

Aerosols are treated as externally mixed and follow a lognormal size distribution, with geometric mean radius and standard deviation values derived from established studies[62]. Specifically, sulfate ($SO_4$) aerosols are assigned a geometric mean radius of 0.05 μm with a geometric standard deviation of 2.0, consistent with previous modeling investigations[63,64]. This corresponds to an effective radius of approximately 0.166 μm. Black carbon (BC) aerosols have a smaller geometric mean radius of 0.0118 μm and the same standard deviation[64], corresponding to an effective radius of approximately 0.039 μm. These aerosol size assumptions are also adopted in the latest versions of the GFDL models, including both ESM4[44] and CM4[65], as

implemented in recent studies[66,67], ensuring consistency with widely used modeling frameworks. The model includes only direct aerosol effects, with optical properties derived from prior work[62,68]. Water uptake is handled differently for each aerosol type: BC is modeled as a dry aerosol with fixed optical properties, while $SO_4$ aerosols' optical properties adjust based on relative humidity. However, because stratospheric relative humidity remains relatively low and stable, $SO_4$ optical depth shows little variability in our simulations.

In this study, a present-day simulation is integrated for 10 years as a control run in GFDL-AM2.5 with the prescribed SST and sea ice concentration of the monthly climatology of the period 1986–2005, along with GHG, aerosol, and ozone concentrations fixed at corresponding levels. To explore the altitude dependence in stratospheric aerosol deployments, we prescribe 0.5 Tg of horizontally uniform aerosols separately within each of the highest 7 sigma levels of the aerosol input file for both BC and $SO_4$. For clearer communication to the public, we convert these sigma levels to approximate pressure levels by multiplying by 1000 hPa and present the corresponding pressure levels in the main text.

In our simulations, we prescribe a constant aerosol mass burden at specific altitudes, maintaining a horizontally uniform distribution throughout the model period. Rather than simulating an injection event followed by transport, mixing, and removal processes, the aerosol layer remains time-invariant, with concentrations fixed at designated sigma levels. This approach simplifies the setup by excluding sedimentation and other dynamic redistributions, which would occur in fully interactive aerosol models. For $SO_4$, we perform an additional set of simulations using 5.0 Tg, which provides a more similar ERF to the 0.5 Tg BC forcing. The two sets of $SO_4$ aerosol simulations can also be used to evaluate the linearity between prescribed $SO_4$ aerosol mass-burdens and corresponding ERF.

The exploration of the altitude dependence in stratospheric aerosol deployments helps to determine the most effective layer for the stratospheric BC aerosol deployment experiments. As demonstrated in the main text, by warming at the altitude where the peak $CO_2$ absorption band achieves unit optical depth, allowing for unimpeded emission to space and therefore making outgoing longwave radiation most sensitive to temperature changes, we are able to maximize the weakening of the $CO_2$ greenhouse effect.

Next, the global coupled climate model (GFDL-CM2.5-FLOR) is used to evaluate the cooling effects of the stratospheric BC aerosol deployment strategy and explore other potential climate impacts of BC deployment relative to the more traditional $SO_4$ deployments (Supplementary Table 1). The GFDL-CM2.5-FLOR is the forecast-oriented low ocean resolution version of GFDL-CM2.5, with a horizontal resolution of ~50 km for the atmosphere and land components developed from GFDL-CM2.5 and a coarser (~1°) resolution for the oceanic and sea ice components from GFDL-CM2.1[69]. Before undertaking any specific investigations, we run a 200-year control simulation by maintaining radiative forcing and land use/land cover at the level of the year 1860. Initializing from one year of the control simulation, 0.5 Tg BC and 5.0 Tg $SO_4$ aerosols are prescribed separately at the most effective layer found in the previous series of fixed-SST simulations and then integrated for 150 years. The cooling effects of the proposed BC intervention strategy are compared with that of the most common solar radiation management with $SO_4$ aerosol deployment. It is worth noting that the high-resolution climate models (GFDL-AM2.5 and GFDL-CM2.5-FLOR) also allow for prospective explorations of the changes in the occurrence of and impacts from extreme weather events (e.g., tropical cyclones and extreme precipitation) to the potential climate interventions.

As an initial assessment of the model dependence within the results presented here and to enable multiple ensemble members for reducing the impact of internal variability, we incorporate a less computationally expensive GFDL model (GFDL-AM2.1 and corresponding coupled model GFDL-CM2.1) for additional corroboration of our results (Supplementary Table 2). This serves two key purposes: first, it aligns with established ERF diagnostic recommendations[13], which suggest 30-year simulations to minimize noise — a more practical approach with AM2.1 due to its lower computational cost. Second, since our multi-ensemble coupled simulations rely on GFDL-CM2.1, which shares the same atmospheric component,

including AM2.1 ensures consistency between atmosphere-only and coupled model experiments. Notably, the aerosol representation in AM2.1 remains identical to that in AM2.5.

## Radiative kernel and approximate partial radiative perturbation methods

To better understand the ERF and the subsequent cooling processes, the radiative kernel method[70] is adopted to decompose the ERF and radiative responses into contributions from each radiatively relevant state variable, including temperature, water vapor, albedo, and cloud. Host-model radiative kernels derived from the atmospheric model GFDL-AM2.1 are used to provide higher accuracy in the decompositions. Here, the radiative kernel method is used primarily for longwave radiation decomposition.

Although the shortwave water vapor absorptions are also calculated using the radiative kernel method, most shortwave decomposition relies on the approximate partial radiative perturbation (APRP) technique. The APRP technique[71] is applied to decompose perturbations in shortwave radiative flux at the TOA into contributions from changes in surface albedo, non-cloud atmospheric constituents, and clouds. We consider the non-cloud contribution to be the sum of the shortwave water vapor absorptions and the direct effects (e.g., absorption and scattering) of aerosol perturbations. Therefore, the direct effects of aerosol perturbations [or the shortwave instantaneous radiative forcing (IRF)] are calculated as the difference between the non-cloud component from the APRP technique and the shortwave water vapor absorptions of the radiative kernel method.

We chose to use the APRP technique for the shortwave decomposition to avoid prescribing the unknown cloud masking extent required for the shortwave IRF in the radiative kernel method decomposition. Aided by the combined radiative kernel and APRP method, the differences in decomposed components, including the IRF and rapid adjustments of the atmosphere initiated by the two aerosol types, will help to better understand their differences in the resultant ERF.

## Data availability

The data and scripts used to produce the figures at Zenodo[72].

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

## Acknowledgements

We thank the three anonymous reviewers for their insightful suggestions and constructive comments, which have greatly improved the manuscript. This work was supported by Award MPS-SRM-00006584 from the Simons Foundation, Award 80NSSC22K0999 from the National Aeronautics and Space Administration, and Award DE-SC0021333 from the United States Department of Energy.

## Author contributions

H.H., B.J.S., and G.A.V. designed research; H.H. and W.Y. performed research; H.H. analyzed data; H.H., B.J.S., G.A.V., and W.Y. contributed to the interpretation of the results; H.H., B.J.S., and G.A.V. wrote the paper.

## Competing interests

The authors declare no competing interests.
