## [Transparent Peer Review file · Communications Earth & Environment]

Stratospheric aerosol injection can weaken the carbon dioxide greenhouse effect

Corresponding Author: Dr Haozhe He

Version 0:

Decision Letter:

Dear Dr He,

Your manuscript titled "Weakening the CO₂ Greenhouse Effect via Stratospheric Aerosol Injection" has now been seen by 3 reviewers, whose comments are appended below. You will see that they find your work is novel and potentially impactful. However, they have raised quite substantial concerns that must be addressed. In light of these comments, we cannot accept the manuscript for publication, but would be interested in considering a revised version that fully addresses these serious concerns.

In particular, we require that you:

- fully justify your modelling approach, in particular towards black carbon
- include discussions on the potential side effects in the stratosphere and troposphere
- fully acknowledge logistical challenges of this method.

We hope you will find the reviewers' comments useful as you decide how to proceed. Should additional work allow you to address these criticisms, we would be happy to look at a substantially revised manuscript. If you choose to take up this option, please either highlight all changes in the manuscript text file, or provide a list of the changes to the manuscript with your responses to the reviewers.

When resubmitting, please provide a point-by-point response to the reviewers' comments. Please submit your responses as a separate file, distinct from your cover letter where you can add responses to the Editors' comments that you do not want to be made available to the reviewers. Word files are preferred. We recommend that any figures, tables or graphs that are included in the response to reviewers are also included in the main article or Supplementary Information.

If the revision process takes significantly longer than three months, we will be happy to reconsider your paper at a later date, as long as nothing similar has been accepted for publication at Communications Earth & Environment or published elsewhere in the meantime.

Please use the following link to submit your revised manuscript, point-by-point response to the reviewers' comments with a list of your changes to the manuscript text (which should be in a separate document to any cover letter), a tracked-changes version of the manuscript (as a PDF file) and any completed checklist:

Link Redacted

Please do not hesitate to contact us if you have any questions or would like to discuss the required revisions further. Thank you for the opportunity to review your work.

Best regards,

Dr Zijun Li
Editorial Board Member
Communications Earth & Environment
orcid.org/0000-0002-2973-1216

Alice Drinkwater, PhD
Associate Editor
Communications Earth & Environment

EDITORIAL POLICIES AND FORMAT

If you decide to resubmit your paper, please ensure that your manuscript complies with our editorial policies and complete and upload the checklist below as a Related Manuscript file type with the revised article:

Editorial Policy Policy requirements
(Download the link to your computer as a PDF.)

- Behavioural and social science
- Ecological, evolutionary & environmental sciences
- Life sciences

<https://www.nature.com/documents/nr-reporting-summary.zip>

For your information, you can find some guidance regarding format requirements summarized on the following checklist: (<https://www.nature.com/documents/commsj-phys-style-formatting-checklist-article.pdf>) and formatting guide (<https://www.nature.com/documents/commsj-phys-style-formatting-guide-accept.pdf>).

REVIEWER COMMENTS:

Reviewer #1 (Remarks to the Author):

The manuscript "Weakening the CO₂ Greenhouse Effect via Stratospheric Aerosol Injection" by He et al. explores the use of stratospheric aerosol injections as a method to mitigate global warming. Unlike previous studies that have suggested injecting reflective aerosols into the lower troposphere, this study proposes injecting absorbing particles into the upper atmosphere. At first, the idea of injecting absorbing aerosols struck me as quite unconventional, as it goes against the commonly held belief about the most effective geoengineering strategies (preferring scattering aerosols, not absorbing), underscoring the originality and novelty of their research. However, the authors demonstrate that this approach could work in theory, which justifies its consideration for publication.

That said, I still have reservations about how this method would perform in practice. There are also some limitations in the modeling techniques used, leaving me uncertain about the study's overall implications. Nevertheless, given the innovative nature of the idea and its potential to spark new discussions within the scientific community, I am inclined to recommend its publication. Before that, a few points require clarification, and the practical challenges and modeling limitations should be explicitly addressed in the manuscript.

The first point of confusion for me relates to the title, "Weakening the CO₂ Greenhouse Effect via Stratospheric Aerosol Injection," which is referenced multiple times throughout the text. Perhaps I'm misunderstanding something, but how exactly do absorbing aerosols weaken the CO₂ greenhouse effect?

I've been trying to conceptualize this method, imagining the upper atmosphere as a separate layer around Earth. This layer absorbs incoming radiation, reducing the amount that reaches the surface, while also emitting longwave (LW) radiation. However, the emitted LW radiation is less than the reduced shortwave (SW) radiation, resulting in a net cooling effect on Earth. Yet, this process seems unrelated to the greenhouse effect or CO₂ specifically, and it appears to me that the method would work even in the absence of CO₂. Am I missing a critical aspect here?

As I mentioned, I also have some concerns about the technical feasibility of this method. Transporting and injecting large quantities of aerosols into the lower stratosphere is no small task. How exactly would this be accomplished at an altitude of >30 km?

The third, and perhaps most significant, concern is the transport of aerosols. If I've understood correctly, this study assumes that aerosols of a specific size are simply distributed within a certain vertical layer of the model. However, in reality, aerosols are transported throughout the atmosphere, and I'm unsure how this would occur at such high altitudes—or whether we even have models capable of accurately simulating it. A critical issue is that aerosols will eventually descend. Once they reach the lower layers of the stratosphere, they could warm the atmosphere without providing a significant cooling effect. Additionally, aerosol microphysics—affecting size, composition, and thus optical properties—would play a crucial role, but this is not taken into account in this study.

As I mentioned, the idea presented in this manuscript is both novel and intriguing, and I believe the manuscript has value simply by introducing this concept. Therefore, I don't see the aforementioned shortcomings as a barrier to publication. However, these aspects should at least be acknowledged and discussed in the study.

Then I have couple of minor comments:

L18: I don't fully understand what is meant by “warming the emission level of CO₂.” Could you clarify?

L36, L42: Most studies suggest the injection of SO₂ gas rather than sulfate aerosol.

L47-48: I'm unclear on the meaning of “where the CO₂ absorption bands achieve unit optical depth.” Could you elaborate?

L60-61: “These studies did not consider the impact of BC in modifying the greenhouse effect.” Related to my earlier question, I don't fully understand the physical mechanism by which BC or the stratospheric temperature affects the greenhouse effect. Could you provide a more detailed explanation?

L98-100: “Air temperature increases within the upper stratosphere provide a more efficient longwave emission to space. Thus, one can reduce the strength of the CO₂ greenhouse effect by reducing the concentration of CO₂ or warming the emission level of CO₂.” I understand that the higher the altitude of radiation emission, the less likely it is to be trapped in the atmosphere. However, I'm not clear about the conclusion made from that (we can use this to reduce the greenhouse effect of CO₂) Could you clarify this?

L109: “...suggesting a roughly linear relationship between SO₄ aerosol loads and the corresponding ERF.” I think this is more or less expected, but it would not hold if aerosol microphysics were taken into account.

L147: “...the CO₂ emission to space” — could you clarify what this means?

L152: What altitude were the aerosols injected at in these coupled simulations? I see this information in Extended Data Table 1, but it might be helpful to mention it here as well.

L167-169: “Rapid precipitation decreases are found in both of these climate intervention simulations (Fig. 3b and Extended Data Fig. 4b). In particular, the fast precipitation decline in the BC simulations occurs without significant surface temperature changes.”

I'm not sure I fully follow this point. The fast precipitation response appears quite small, as the intercept in Extended Data Fig. 4b is 0.01 mm for both cases. You suggest that the response would be larger for BC, but the figure doesn't seem to support this. That said, I was expecting to see a larger reduction in precipitation given the linear dependence between absorbed radiation and fast precipitation response (as shown in Samset et al., 2016, and Laakso et al., 2024). This relationship seems to hold for absorbing aerosols in the troposphere or lower stratosphere, based on earlier studies, but perhaps the dynamics differ when aerosols are placed at the highest levels of the model. Could you clarify this?

Samset, B. H., et al., *Geophys. Res. Lett.*, 43, 2782–2791, <https://doi.org/10.1002/2016GL068064>, 2016

Laakso, A., et al., *Earth Syst. Dynam.*, 15, 405–427, <https://doi.org/10.5194/esd-15-405-2024>, 2024.

Related to Extended Data Fig. 4a: The relationship between TOA energy imbalance and surface temperature appears quite scattered, whereas it typically shows a clearer linear response. However, given the relatively small response here compared to, for example, 4xCO₂ simulations, this might simply be normal noise. Do you think this is expected behavior in GFDL-CM2.1, or could it be a result of this unusual case, where model perturbations are applied at the uppermost levels?

Fig. 4: This is a minor point, but I would suggest using a different map projection to better highlight differences at higher latitudes and near the Prime Meridian.

L227: A small detail too, but there wasn't an actual “injection” in these simulations. As mentioned in the next line, the study focused on the effects of different aerosol layer locations. Results could differ significantly if aerosol injection and transport processes were explicitly simulated.

L279-L287: How are aerosols represented in the model? Do they have an assumed fixed size? Is any water uptake considered, or are these treated as dry aerosols?

L297: “0.5 Tg BC and 5.0 Tg SO₄ aerosols are prescribed separately at the most effective layer found in the previous series of fixed-SST simulations.” According to Extended Data Table 1, the aerosols are placed at the second-highest level. However, based on GFDL-CM2 results, these are not the levels where the ERF is maximized. Could you clarify?

L307: I'm not entirely sure I understand the purpose of including the GFDL-AM2.1 model here. While using an additional model is certainly always valuable, did it provide any specific insights or added value beyond what was obtained from GFDL-AM2.5?

Vertical levels have been referred to as sigma levels in several figures. I would prefer to use pressure or altitude levels/units as they are easier to understand.

Reviewer #2 (Remarks to the Author):

This manuscript presents a new approach to stratospheric aerosol injection (SAI) for climate intervention by introducing absorptive aerosols, particularly black carbon (BC), in the upper stratosphere around the 10 hPa level (near the emission level of CO₂). The study hypothesizes that this strategy could weaken CO₂'s greenhouse effect by increasing outgoing longwave radiation at the top of the atmosphere, offering potentially ~12x more efficient cooling than traditional reflective

aerosols injected into the lower-mid stratosphere. Global model simulations explore the radiative effects of this method, highlighting potential benefits and calling for further research on unintended impacts. The paper aligns with the scope of Communications Earth & Environment and would likely interest the broader atmospheric and climate science community.

However, I have several major concerns and questions regarding the analysis and conclusions:

1. The air density at the 10 hPa level is extremely low, which challenges the long-term suspension of aerosols, compared to traditional SAI proposals targeting the lower-mid stratosphere. How long can BC remain at these high altitudes? Assessing particle lifetimes at this altitude and considering logistical requirements would strengthen the study's feasibility assessment.
2. The stratospheric budget of aerosols over time is not clearly explained in the paper. Specifically, do the authors prescribe constant aerosol amounts (e.g., 0.5 Tg or 5 Tg) at specific altitudes throughout the model period, or do they inject aerosols only at the beginning of the model runs? It seems that the former approach is used, but if so, BC injected at 10 hPa would sediment through the lower and middle atmosphere, impacting layers far below the target altitude. This sedimentation process, which appears underrepresented in the model, could lead to unintended warming of the lower stratosphere and tropopause, water vapor feedbacks, and ozone impacts. More importantly, this could significantly affect the model's cooling efficiency. Therefore, it is essential for the authors to explicitly describe the aerosol injection method used in their simulations and analyze the consequences of BC aerosol sedimentation.
3. Given the likely path of BC aerosols through the Brewer-Dobson circulation, much of the injected BC may eventually sediment at the surface in high-altitude or polar regions. This could significantly alter the surface albedo of ice and snow, potentially offsetting the cooling effects and causing a positive feedback. This limitation should also be discussed.
4. The warming of the upper stratosphere induced by BC aerosols could alter stratospheric dynamics, such as changes in the Brewer-Dobson circulation and polar vortex behavior. Additionally, BC interactions with stratospheric ozone could exacerbate ozone depletion, leading to higher surface UV radiation and potentially counteracting any cooling effects. Moreover, warming in the upper stratosphere could increase water vapor, especially if tropopause temperatures rise. These effects could diminish or offset the cooling by adding additional greenhouse gases. Were these potential side effects considered in the model simulations?
5. Simulations use a pre-industrial baseline, which, while simplifying the analysis, does not reflect modern greenhouse gas concentrations, which are the primary motivation for the study. Given that weakening CO₂ forcing is a central aim, it would be more informative to model the intervention under current or projected CO₂ levels to provide a realistic view of its effectiveness in today's atmospheric condition.

I also have some specific comments:

1. Line 54-55: Please provide relevant references for "analytic and radiative transfer calculations demonstrate the feasibility of weakening the CO₂ greenhouse effect."
2. Line 114: Please explain what "sigma level" means and why this coordinate system is beneficial over pressure levels for a more general audience.
3. Figure 1 and Figure 4: Please add units to the color bars.

Reviewer #3 (Remarks to the Author):

In this paper, the authors are trying to show that the addition of black carbon in the mid-upper stratosphere (10 hPa) has a much stronger radiative effect than sulfate, which is normally used for stratospheric aerosol injection (SAI) simulations: they explain that this difference is driven by a strong LW effect by the absorbing aerosols in the case of BC.

While in general the manuscript is well written and the results interesting, there is a major flaw at this point that prevents me from understanding the results fully, and that the authors will need to address before moving further.

Essentially, the description of the methods is not in depth enough to allow me to understand what exactly has been done in the simulations. The authors often use the word "prescribed" when referring to the aerosol layer. First, I'd like more details about this. Did you prescribe an optical depth uniformly? Can one see it? How does it compare to other distributions used in other papers?

Secondly, I think -but I can only guess, not having seen the actual details- that, if the authors have prescribed the aerosols, they are missing out on the potential very, very relevant effect of aerosol lifetime in their simulations. This is important because they name their simulations based on mass (0.5 or 5 Tg), and then claim that "that the reduction in global temperatures induced through this process is an order of magnitude larger (per unit aerosol mass) than the injection of more traditional reflective aerosols". But you can't make a unit mass comparison without including considerations about lifetime, and we know that, for SO₂ injections, lifetime matters greatly!

See for instance Lee et al. (2023). Therefore, the authors should re-frame their comparison in terms of optical depth, not mass, as a prescribed mass doesn't say much about the amount of material that would need to be injected in order to achieve that layer (and that is the mass that matters!). Again, I think I would be able to have more meaningful suggestions once the details have been cleared out by the authors. These details must also include information about the size distribution of the injected aerosols, which is very relevant information. What is the effective radius for both aerosol populations?

It is also important that the authors don't simply dismiss the fact that the BC simulations have a upper stratospheric warming of 30-50 K (!!). In the conclusions, this, coupled with the acknowledgment that their models lack proper stratospheric dynamics (due to a low top I assume) and stratospheric chemistry, should really dampen the certainty the authors use also in the abstract about how effective BC would be. I assume they use prescribed ozone levels in the stratosphere, and it would be useful to discuss how a potentially very high ozone depletion would affect this, and how stratospheric circulation would change. I look forward to see these details, and I hold on judgment about how publishable these results are until then.

Lastly, I would suggest the authors acknowledge more recent studies in fully coupled models using BC, such as Haywood et al. (2022)

Haywood, J. M., Jones, A., Johnson, B. T., and McFarlane Smith, W.: Assessing the consequences of including aerosol absorption in potential stratospheric aerosol injection climate intervention strategies, *Atmos. Chem. Phys.*, 22, 6135–6150, <https://doi.org/10.5194/acp-22-6135-2022>, 2022.

Lee, W. R., Visoni, D., Bednarz, E. M., MacMartin, D. G., Kravitz, B., & Tilmes, S. (2023). Quantifying the efficiency of stratospheric aerosol geoengineering at different altitudes. *Geophysical Research Letters*, 50, e2023GL104417. <https://doi.org/10.1029/2023GL104417>

Communications Earth & Environment is committed to improving transparency in authorship. As part of our efforts in this direction, we are now requesting that all authors identified as 'corresponding author' create and link their Open Researcher and Contributor Identifier (ORCID) with their account on the Manuscript Tracking System prior to acceptance. ORCID helps the scientific community achieve unambiguous attribution of all scholarly contributions. You can create and link your ORCID from the home page of the Manuscript Tracking System by clicking on 'Modify my Springer Nature account' and following the instructions in the link below. Please also inform all co-authors that they can add their ORCIDs to their accounts and that they must do so prior to acceptance.

Version 1:

Decision Letter:

Dear Dr He,

Your manuscript titled "Weakening the CO2 Greenhouse Effect via Stratospheric Aerosol Injection" has now been seen by our reviewers, whose comments appear below. In light of their advice we are delighted to say that we are happy, in principle, to publish a suitably revised version in *Communications Earth & Environment*.

We therefore invite you to revise your paper one last time to address the remaining concerns of our reviewers, which pertain to specifying that the model simulations were conducted with an idealized model and discussing the input size distributions in the model. At the same time we ask that you edit your manuscript to comply with our format requirements and to maximise the accessibility and therefore the impact of your work.

EDITORIAL REQUESTS:

****Please take care to match our formatting and policy requirements. We will check revised manuscript and return manuscripts that do not comply. Such requests will lead to delays. ****

SUBMISSION INFORMATION:

OPEN ACCESS:

Communications Earth & Environment is a fully open access journal. Articles are made freely accessible on publication. For further information about article processing charges, open access funding, and advice and support from Nature Research, please visit <https://www.nature.com/commsenv/open-access>

Link Redacted

Best regards,

Alice Drinkwater, PhD
Associate Editor
Communications Earth & Environment
Consulting Editor
Communications Sustainability

REVIEWERS' COMMENTS:

Reviewer #1 (Remarks to the Author):

After initially reading and reviewing the manuscript "Weakening the CO₂ Greenhouse Effect via Stratospheric Aerosol Injection" by He et al., I had some questions about the design of the study and the limitations of the modeling approach. Overall, I think my concerns were addressed quite well.

My first concern was with the title and the idea of "weakening the CO₂ greenhouse effect". I was initially unsure whether the proposed method of injecting absorbing aerosols into the upper atmosphere and the resulting radiative effects were directly related to the greenhouse effect of CO₂. The authors addressed this issue by repeating their analysis using offline radiative transfer calculations with CO₂ removed, demonstrating that CO₂ does indeed play a major role in the results. This was a valuable addition to the manuscript.

I also had questions about the feasibility of aerosol injection at higher altitudes, as is typically proposed, and the limitations inherent in the aerosol modeling approach. The authors have now acknowledged these limitations in the revised manuscript. They have also responded to my questions in detail, supported their answers with figures, and clarified several points in the text.

Overall, despite some limitations in the modeling approach (which the authors have appropriately acknowledged), I believe the manuscript is original and novel, presents interesting results, and is well written and thorough. I recommend it for publication.

Reviewer #2 (Remarks to the Author):

I appreciate the authors' substantial efforts to address my comments in my initial review, particularly regarding the technical and scientific limitations of this study. The manuscript now includes a more detailed description and discussion of uncertainties and overlooked factors related to aerosol transport, microphysical evolution, sedimentation, chemical and dynamical perturbations in the stratosphere, and the technical challenges of high-altitude aerosol deployment.

However, I remain concerned that these important caveats are not sufficiently reflected in the framing of the manuscript's main conclusions. The Abstract, Introduction, and Conclusion continue to strongly emphasize the potential advantages of the proposed upper-stratospheric absorbing aerosol method, including claims of "an order of magnitude greater efficiency"

compared to traditional sulfate SAI, without clearly communicating that these results stem from highly idealized model experiments that do not include interactive aerosol sedimentation, dynamics, chemistry, or real-world operational considerations.

In my view, the current presentation could lead readers to overinterpret the practical viability and comparative advantage of the method. To prevent this, I recommend that the authors revise the manuscript to more explicitly qualify their main claims throughout, especially in the Abstract and Conclusion, making clear that the reported efficiency gains are specific to an "idealized model setup" and are subject to substantial scientific and technical uncertainties.

Provided that these revisions are made, i.e. clarifying the scope and limitations of the study in the main messaging (Abstract, Introduction, and Conclusion) to avoid overinterpretation of the analyses, I would be supportive of publication. I believe this work would then represent a valuable contribution to the ongoing exploration of SAI strategies and would help spark valuable discussion within the community.

Reviewer #3 (Remarks to the Author):

I appreciate the detailed and thoughtful responses from the authors. I think they did an excellent job expanding the manuscript from its previous form to better highlight the limitations of their study.

However, I need to make one last note based on the expanded explanations: the size distributions for the bulk aerosol parametrizations that they use are incredibly small. 0.05 μm for sulfate and 0.01 μm for BC are very, very low values that would give rather different results in terms of TOA forcing compared to more "realistic" size distributions.

Mind you, this is not just pickiness: BC in nature is usually a much larger size (for instance, the size distribution from Australian fires was bimodal with 0.1 micron mean radius for the fine mode (one oom larger than the ones used here) and most of the mass in a second mode with mean radius of 5 μm (see Yang et al., 2021). For sulfate, most of the mass would also be in accumulation/coarse mode (see Laakso et al., 2022). Longwave absorption is heavily dependant on this (see Dykema et al., 2016), therefore it is unlikely that the authors' results could ever be replicated in other models with any kind of interactive aerosol parametrizations.

Yang, X., Zhao, C., Yang, Y., Yan, X., and Fan, H.: Statistical aerosol properties associated with fire events from 2002 to 2019 and a case analysis in 2019 over Australia, *Atmos. Chem. Phys.*, 21, 3833–3853, <https://doi.org/10.5194/acp-21-3833-2021>, 2021.

Laakso, A., Niemeier, U., Vioni, D., Tilmes, S., and Kokkola, H.: Dependency of the impacts of geoengineering on the stratospheric sulfur injection strategy – Part 1: Intercomparison of modal and sectional aerosol modules, *Atmos. Chem. Phys.*, 22, 93–118, <https://doi.org/10.5194/acp-22-93-2022>, 2022.

Dykema, J. A., D. W. Keith, and F. N. Keutsch (2016), Improved aerosol radiative properties as a foundation for solar geoengineering risk assessment, *Geophys. Res. Lett.*, 43, 7758–7766, doi:10.1002/2016GL069258.

We thank three anonymous reviewers for their insightful suggestions and constructive comments, which significantly improved the manuscript. Before moving on to the specific responses to each reviewer's comments, we would like to clarify a small change we have made, now reflected in the updated Extended Data Table 2.

An examination of our pre-industrial control run using GFDL-CM2.1 revealed a ~ 0.5 K temperature drop during the first two years before reaching equilibrium. This arose from substituting the original aerosol input file with the one used in GFDL-AM2.5 and GFDL-CM2.5-FLOR to ensure consistency between simulations conducted by the two model groups. While this introduced a minor impact on the climate intervention simulations branching from year 1 of the control run, we addressed this by initiating a new ensemble branching from year 11. Importantly, the overall results showed no noticeable changes, confirming the robustness of our conclusions.

REVIEWER COMMENTS:

Reviewer #1 (Remarks to the Author):

The manuscript "Weakening the CO₂ Greenhouse Effect via Stratospheric Aerosol Injection" by He et al. explores the use of stratospheric aerosol injections as a method to mitigate global warming. Unlike previous studies that have suggested injecting reflective aerosols into the lower troposphere, this study proposes injecting absorbing particles into the upper atmosphere. At first, the idea of injecting absorbing aerosols struck me as quite unconventional, as it goes against the commonly held belief about the most effective geoengineering strategies (preferring scattering aerosols, not absorbing), underscoring the originality and novelty of their research. However, the authors demonstrate that this approach could work in theory, which justifies its consideration for publication.

That said, I still have reservations about how this method would perform in practice. There are also some limitations in the modeling techniques used, leaving me uncertain about the study's overall implications. Nevertheless, given the innovative nature of the idea and its potential to spark new discussions within the scientific community, I am inclined to recommend its publication. Before that, a few points require clarification, and the practical challenges and modeling limitations should be explicitly addressed in the manuscript.

Response: We thank the reviewer for the thoughtful and constructive feedback that helped us further advance the manuscript. We appreciate your recognition of the novelty of our approach and its potential to contribute to the ongoing discussion on climate intervention strategies. We also acknowledge the concerns you raise regarding practical implementation and modeling limitations. We have carefully addressed these points in the revised manuscript to provide a clearer assessment of both the feasibility and challenges associated with this method. We address individual comments below.

The first point of confusion for me relates to the title, "Weakening the CO₂ Greenhouse Effect via Stratospheric Aerosol Injection," which is referenced multiple times throughout the text. Perhaps I'm misunderstanding something, but how exactly do absorbing aerosols weaken the CO₂ greenhouse effect? I've been trying to conceptualize this method, imagining the upper atmosphere as a separate layer around Earth. This layer absorbs incoming radiation, reducing the amount that reaches the surface, while also emitting longwave (LW) radiation. However, the emitted LW radiation is less than the reduced shortwave (SW) radiation, resulting in a net cooling effect on Earth. Yet, this process seems unrelated to the greenhouse effect or CO₂ specifically, and it appears to me that the method would work even in the absence of CO₂. Am I missing a critical aspect here?

Response: Thank you for the comment! Your conceptualization is largely correct, especially from a surface or tropospheric perspective. However, our approach focuses on effective radiative forcing (ERF) at the top of atmosphere (TOA), which provides a more comprehensive representation of the overall climate impact of a forcing agent (or most effectively drives surface temperature changes), as it accounts for all atmospheric adjustments (e.g., Boucher et al., 2013; Forster et al., 2016, 2021; Myhre et al., 2013; Smith et al., 2018, 2020).

The key mechanism behind our approach is that absorbing aerosols heat the upper stratosphere by absorbing incoming solar radiation. Since the upper stratosphere is highly effective at emitting

longwave radiation back to space, where the CO₂ absorption bands achieve unit optical depth, allowing for unimpeded emission to space and therefore making outgoing longwave radiation most sensitive to temperature changes at this level, this warming results in a net negative radiative forcing at TOA, which in turn leads to surface cooling. The crucial link to the CO₂ greenhouse effect lies in the fact that CO₂ forcing strongly depends on the temperature of the upper stratosphere (He et al., 2023; Jeevanjee et al., 2021). When CO₂ concentrations increase, they enhance the greenhouse effect by trapping outgoing longwave radiation. However, if the upper stratosphere is heated, it increases the emission of longwave radiation to space, effectively counteracting part of the CO₂-induced warming.

To clarify your question about whether this method would work in the absence of CO₂: without CO₂, the atmosphere is optically thinner in the infrared, meaning terrestrial radiation can escape to space with minimal absorption or scattering. As a result, temperature changes in the stratosphere would have little influence on longwave radiation fluxes at TOA. In fact, our offline radiative transfer calculations show that, in a CO₂-free atmosphere, an identical temperature perturbation in the upper stratosphere would induce only about one-tenth of the radiative change observed in a CO₂-rich atmosphere (Figure R1). This highlights the fundamental role of CO₂ in modulating the climate response to upper-stratospheric heating.

Figure R1. The net downward longwave radiative flux sensitivities at the TOA to per K temperature change ($W m^{-2} K^{-1} / 100 hPa$) at individual pressure layers (hybrid-sigma level) obtained from a series of temperature perturbations ranging from 1 K to 50 K. The negative sensitivities to per K warming indicate more infrared radiation emitted out of the TOA by the stratospheric warming. Corresponding layer thicknesses normalize the sensitivity results and thereby represent the response to uniform perturbations in 100-hPa-thick layers. The results are shown from offline radiative flux calculation (a) with CO₂ and (b) without CO₂.

Correspondingly, the clarification has also been added in the manuscript as follows:

“We show that deploying absorptive aerosols in the upper stratosphere (~10 hPa) increases the emission of infrared radiation at the top-of-atmosphere.”

“Admittedly, a few previous studies of SAI using BC aerosols have been performed (Ferraro et al., 2011; Jones et al., 2016; Kravitz et al., 2012); however, these studies did not consider the above-mentioned impact of BC in modifying the greenhouse effect, by altering the thermal structure of the upper stratosphere, focusing instead on its ability to cool the planet by reducing the solar radiation at the surface, either through radiative forcing at the tropopause or surface level. However, effective radiative forcing (ERF) at the TOA provides a more

comprehensive representation of the overall climate impact of a forcing agent, as it accounts for all atmospheric adjustments (e.g., Boucher et al., 2013; Forster et al., 2016, 2021; Myhre et al., 2013; Smith et al., 2018, 2020)."

"The atmospheric greenhouse effect fundamentally depends on two quantities: the concentration of GHGs and the atmospheric temperature structure. An analytical model demonstrates that the radiative heat trapped by CO₂ represents a swap of tropospheric emission for stratospheric emission and, therefore, can be understood in terms of a dependence on the emission temperature of both the stratosphere and troposphere (Jeevanjee et al., 2021). The former refers to the temperature of the upper stratosphere, where the CO₂ absorption bands achieve unit optical depth, allowing for unimpeded emission to space and therefore making outgoing longwave radiation most sensitive to temperature changes, while the latter depends on surface temperature and free-troposphere relative humidity. Recently, He et al. (2023) used this model in conjunction with detailed radiative transfer calculations to demonstrate the dominant role that changes in upper stratospheric temperature play in determining the magnitude of the radiative forcing and greenhouse effect of CO₂.

Figure 1 shows the sensitivity of the outgoing longwave radiation to the temperature perturbations within the stratosphere ranging from 100 hPa to 1 hPa. The results are obtained from a set of offline radiative transfer calculations using monthly climatological temperature and humidity profiles of pre-industrial simulations as well as the specified temperature perturbations at individual layers, by a broadband radiative transfer model [SOCRATES (Edwards & Slingo, 1996; Manners et al., 2015)].

These results demonstrate the importance of the vertical location of warming on the emission of outgoing longwave radiation. For the same magnitude of perturbation, a change in temperature at a higher altitude has a larger impact on the outgoing longwave radiation. Optimal sensitivity occurs when the temperature perturbation coincides with the level where CO₂ optical depth is ~1 (upper stratosphere), allowing for unimpeded emission to space. Air temperature increases within the upper stratosphere provide a more efficient longwave emission to space. Thus, one can reduce the strength of the CO₂ greenhouse effect by reducing the concentration of CO₂ or warming the emission level of CO₂. Note that the same temperature perturbation in a CO₂-free atmosphere would result in only one-tenth of the outgoing longwave radiation change (Supplementary Figure 1). Therefore, most of the changes in outgoing longwave radiation shown in Figure 1 are due to the modulation of the CO₂ greenhouse effect."

This clarification also addresses two identical minor comments raised by the reviewer in the following.

L60-61: "These studies did not consider the impact of BC in modifying the greenhouse effect." Related to my earlier question, I don't fully understand the physical mechanism by which BC or the stratospheric temperature affects the greenhouse effect. Could you provide a more detailed explanation?

Response: Thank you for the specific comment. Please refer to the previous response, which addresses this point. We have also clarified this in the updated manuscript, as shown above.

L98-100: "Air temperature increases within the upper stratosphere provide a more efficient longwave emission to space. Thus, one can reduce the strength of the CO₂ greenhouse effect by reducing the concentration of CO₂ or warming the emission level of CO₂." I understand that the

higher the altitude of radiation emission, the less likely it is to be trapped in the atmosphere. However, I'm not clear about the conclusion made from that (we can use this to reduce the greenhouse effect of CO₂) Could you clarify this?

Response: Thank you for the specific comment. Please refer to the previous response, which addresses this point. We have also clarified this in the updated manuscript, as shown above.

As I mentioned, I also have some concerns about the technical feasibility of this method. Transporting and injecting large quantities of aerosols into the lower stratosphere is no small task. How exactly would this be accomplished at an altitude of >30 km?

Response: We appreciate the reviewer's concern regarding the technical feasibility of aerosol injection at altitudes exceeding 30 km. While large-scale implementation presents substantial engineering challenges, multiple studies have explored plausible delivery mechanisms, and some existing technologies could potentially be adapted for this purpose.

Aerosol injection into the lower and upper stratosphere could be achieved through several proposed methods (National Academies of Sciences, Engineering, and Medicine, 2021):

1. High-Altitude Aircraft: Modified versions of high-altitude aircraft, such as the Stratospheric Aerosol Injection Lofter concept, have been investigated for their capability to reach ~20 km. While current aircraft struggle to operate beyond this altitude, advancements in aerodynamics, propulsion, and payload capacity could enable them to reach higher levels.
2. Stratospheric Balloons: Tethered or free-floating balloons could serve as a potential means of delivering aerosol payloads to high altitudes. However, precise control over dispersal and the challenge of scaling such operations remain unresolved.
3. High-Altitude Rockets or Artillery: While more speculative, launching aerosols via high-altitude rockets or artillery has been explored in geoengineering literature as a means to reach altitudes well above 30 km. This approach could be feasible for targeted, high-altitude deployments but may face cost and reusability constraints.
4. Hybrid Approaches: Combining multiple delivery methods—such as using aircraft to carry aerosol payloads that are subsequently lofted by balloon or small rocket propulsion—could help overcome the altitude limitations of individual methods.

Beyond direct delivery mechanisms, the self-lofting effect of absorbing aerosols (e.g., Gao et al., 2021; Haywood et al., 2022) could also play a role in maintaining aerosols at higher altitudes. This phenomenon occurs when aerosols absorb solar radiation, heat the surrounding air, and induce upward motion, which may help sustain their presence in the upper stratosphere after initial injection.

Generally, several feasibility assessments (e.g., McClellan et al., 2012; Smith & Wagner, 2018) suggest that while delivery to 30 km remains a significant challenge, it is not necessarily infeasible with continued technological advancements. The development of a fleet of specialized aircraft capable of reaching at least 20 km already appears feasible with current technology, and further engineering innovations could extend these capabilities. However, we acknowledge that scaling, cost, and logistical challenges remain key uncertainties that warrant further investigation.

We appreciate this important question and have discussed these feasibility considerations in the manuscript as following.

“Admittedly, delivering aerosols to the upper stratosphere presents significant engineering challenges (National Academies of Sciences, Engineering, and Medicine, 2021). While feasibility assessments (e.g., McClellan et al., 2012; Smith & Wagner, 2018) suggest that reaching altitudes around 30 km remains difficult, continued technological advancements may render this more achievable. Current designs for specialized high-altitude aircraft indicate that deployment to around 20 km is already within reach, and further innovations could extend this range. Other options for delivering material to the stratosphere include rockets, ballistic payloads launched from the ground (e.g., artillery, rail guns), and balloons. In addition to direct delivery methods, the self-lofting effect of absorbing aerosols, particularly BC aerosols (e.g., Gao et al., 2021), offers a potential mechanism to sustain aerosols at higher altitudes. This effect relies on the aerosols absorbing solar radiation, heating the surrounding air, and generating upward motion, helping to maintain the particles aloft. However, the efficiency and duration of this process remain uncertain, particularly when factoring in seasonal variations in solar radiation and potential interactions with stratospheric dynamics (Haywood et al., 2022). Moreover, large-scale deployment raises critical concerns about scaling, cost, and logistics. Maintaining a steady, controlled aerosol layer would likely require continuous delivery, adaptive flight strategies, and rigorous monitoring to account for atmospheric variability and aerosol dispersion. These operational challenges, coupled with the need for specialized infrastructure and international coordination, underscore the importance of further research into both the physical science and engineering aspects of stratospheric aerosol delivery.”

The third, and perhaps most significant, concern is the transport of aerosols. If I’ve understood correctly, this study assumes that aerosols of a specific size are simply distributed within a certain vertical layer of the model. However, in reality, aerosols are transported throughout the atmosphere, and I’m unsure how this would occur at such high altitudes—or whether we even have models capable of accurately simulating it. A critical issue is that aerosols will eventually descend. Once they reach the lower layers of the stratosphere, they could warm the atmosphere without providing a significant cooling effect. Additionally, aerosol microphysics—affecting size, composition, and thus optical properties—would play a crucial role, but this is not taken into account in this study.

Response: Thank you for the comment. You raise an important concern regarding the transport and fate of aerosols in the stratosphere, as well as the limitations of our current modeling approach.

In our simulations, we prescribe a fixed aerosol distribution at specific altitudes throughout the model period, rather than simulating an injection event followed by subsequent transport, microphysical evolution, and removal processes. This means that aerosols of a specific size are distributed within a designated vertical layer in our model, and their concentrations remain time-invariant. We have clarified the simulation setup in the Materials and Methods section as follows:

“In our simulations, we prescribe a constant aerosol mass burden at specific altitudes, maintaining a horizontally uniform distribution throughout the model period. Rather than simulating an injection event followed by transport, mixing, and removal processes, the aerosol layer remains time-invariant, with concentrations fixed at designated sigma levels. This approach simplifies the setup by excluding sedimentation and other dynamic redistributions, which would occur in fully interactive aerosol models.”

While this approach simplifies the analysis and allows us to isolate the radiative effects of upper-stratospheric BC aerosols, we acknowledge that it does not capture key physical processes such as sedimentation, transport by the Brewer-Dobson Circulation (BDC), and microphysical changes in aerosol size and composition over time. If BC aerosols were injected at ~ 10 hPa in a fully interactive model, we would expect them to gradually descend through the stratosphere, leading to several potentially important consequences:

1. **Efficiency Reduction & Climate Impacts:** As BC aerosols move downward, they could warm the lower stratosphere, reducing the net cooling efficiency of the intervention. This heating could also alter temperature gradients, affecting stratospheric stability and potentially impacting tropospheric circulation patterns.
2. **Water Vapor Feedbacks:** BC sedimentation could enhance lower-stratospheric heating, potentially increasing water vapor concentrations. Since water vapor is a potent greenhouse gas, such increases could counteract some of the intended cooling effects of BC injection.
3. **Seasonal and Latitudinal Dependence:** The self-lofting effect of BC, which arises from solar absorption and heating-induced vertical motion, may play a role in modulating aerosol distribution. This effect is expected to be stronger in summer when sunlight is available and weaker in winter, leading to enhanced downward transport during the colder months. As a result, the seasonal cycle could influence BC residence time and its overall radiative impact.
4. **Ozone Depletion:** Increased lower-stratospheric heating and water vapor changes could exacerbate ozone depletion, particularly in polar regions. While our study uses prescribed ozone levels, previous research suggests that BC-induced stratospheric warming and changes in chemical reactions could accelerate ozone loss.
5. **Geographical Distribution of Effects:** The BDC likely plays a crucial role in redistributing BC aerosols, with most sedimentation occurring at higher latitudes due to subsiding air masses. This could lead to regional variations in stratospheric heating, with potential implications for climate dynamics and high-latitude weather patterns.

We recognize that our simplified setup does not account for these transport and sedimentation processes, and we have clarified this limitation in the revised manuscript. Additionally, our model does not incorporate aerosol microphysics, meaning that factors such as particle size evolution, composition changes, and their effects on optical properties are not explicitly considered. This is an important limitation, as aerosol microphysics can significantly influence both radiative forcing and atmospheric lifetime. Future studies incorporating dynamic aerosol transport, lofting, and microphysical processes will be essential for a more accurate assessment of the feasibility and climate impacts of BC-based stratospheric interventions. We appreciate this important question and have expanded the manuscript discussion accordingly.

“We acknowledge that key uncertainties remain in the transport, microphysical evolution, and sedimentation of absorbing aerosols. Our simulations prescribe aerosol concentrations uniformly at fixed altitudes, neglecting redistribution due to transport processes such as interactions with stratospheric circulation, as well as self-lofting and its seasonal variability driven by changes in incoming solar radiation. Since the BDC plays a crucial role in aerosol transport, anomalous radiative heating could further shape aerosol distribution, modulating their lifetime and climatic impacts (e.g., Lee et al., 2023; Tilmes et al., 2017, 2018). Additionally, aerosol microphysics—governing size and compositional changes through nucleation, condensation,

coagulation, hygroscopic growth, and sedimentation—can significantly affect radiative properties and further influence residence time at the intended altitude (Dykema et al., 2016; Kleinschmitt et al., 2018; Tilmes et al., 2017). As BC aerosols move downward, they may warm the lower stratosphere, reducing the cooling efficiency of the intervention. This warming could also alter temperature gradients, destabilizing the stratosphere and potentially affecting tropospheric circulation patterns. Moreover, the enhanced lower-stratospheric heating may increase water vapor concentrations, which could counteract the intended cooling effects by amplifying the greenhouse effect and exacerbating ozone depletion. In addition, when BC aerosols sediment onto snow and ice surfaces, they reduce surface albedo, causing the snow and ice to absorb more solar radiation rather than reflecting it. This effect accelerates melting, increases surface temperatures, and triggers feedback mechanisms that could offset some of the cooling effects of BC aerosols.”

As I mentioned, the idea presented in this manuscript is both novel and intriguing, and I believe the manuscript has value simply by introducing this concept. Therefore, I don't see the aforementioned shortcomings as a barrier to publication. However, these aspects should at least be acknowledged and discussed in the study.

Response: We appreciate the reviewer's recognition of the novelty of our approach and its potential to advance the discussion on climate intervention strategies. We agree that the limitations of our results, particularly due to the use of a relatively less complex model, should be acknowledged. We have revised the manuscript to clarify our simulation setup and explicitly discuss these limitations.

Then I have couple of minor comments:

L18: I don't fully understand what is meant by “warming the emission level of CO₂.” Could you clarify?

Response: Thank you for the clarifying question — it prompted a clearer explanation. The phrase “warming the emission level of CO₂” refers to temperature changes occurring at the altitude where Earth's outgoing longwave radiation (OLR) is most effectively emitted to space in the spectral regions influenced by CO₂. This altitude corresponds to the unit optical depth level of CO₂, meaning the point where the atmosphere becomes sufficiently transparent for radiation to escape with minimal absorption or re-emission by the layers above. Essentially, temperature changes at this level directly influence the total outgoing longwave radiative flux, as the emitted radiation can leave the atmosphere with little interference.

In the context of CO₂-driven radiative forcing, increasing CO₂ concentrations shift the effective emission level to a higher, relatively colder altitude in the atmosphere, often in the stratosphere. Since higher altitudes in the stratosphere are generally colder than those in the relatively lower troposphere, this shift reduces the efficiency of radiative cooling to space, leading to a net accumulation of energy and, as a result, surface warming.

Thus, when we refer to “warming the emission level of CO₂,” we are talking about how temperature changes at this critical altitude influence the total outgoing longwave radiative flux. As the effective emission level for CO₂-sensitive wavelengths plays a key role in Earth's energy balance, any warming at this level can significantly impact both radiative equilibrium and the overall climate response to changes in greenhouse gas concentrations.

We appreciate this important question and have clarified it in the manuscript as following.

“Warming the emission level of CO₂, where it reaches unit optical depth and therefore outgoing longwave radiation is most sensitive to its temperature changes, weakens the greenhouse effect by altering the thermal structure of the upper stratosphere rather than the concentration of greenhouse gases.”

This clarification also addresses two identical minor comments raised by the reviewer in the following.

L47-48: I’m unclear on the meaning of “where the CO₂ absorption bands achieve unit optical depth.” Could you elaborate?

Response: Thank you for the specific comment. Please refer to the previous response, which addresses this point. We have also clarified this in the updated manuscript as follows:

“This strategy is motivated by theoretical and modeling evidence that the CO₂ greenhouse effect depends strongly on the upper stratospheric temperature (He et al., 2023; Jeevanjee et al., 2021), where the CO₂ absorption bands achieve unit optical depth, allowing for unimpeded emission to space and therefore making outgoing longwave radiation most sensitive to temperature changes at this level.”

L147: “...the CO₂ emission to space” — could you clarify what this means?

Response: Thank you for the specific comment. Please refer to the previous response, which addresses this point. We have also clarified this in the updated manuscript as follows:

“While the temperature dependence of the CO₂ infrared emission to space is based on fundamental physics and robust across models (He et al., 2023; Jeevanjee et al., 2021), cloud adjustments to anthropogenic forcing are poorly understood and differ significantly between models (e.g., Smith et al., 2018, 2020).”

L36, L42: Most studies suggest the injection of SO₂ gas rather than sulfate aerosol.

Response: Thank you for the comment. We have revised the manuscript to clarify this point, acknowledging that most studies focus on SO₂ injections — often using volcanic eruptions as a natural analog — where the gas oxidizes to form sulfate (SO₄) aerosols. The revised text now reads:

“Sulfur dioxide (SO₂) has been featured prominently in SAI research, with volcanic eruptions serving as a natural analog (e.g., Kravitz et al., 2015). During such eruptions, millions of metric tons of SO₂ are injected into the stratosphere, where they oxidize to form sulfate (SO₄) aerosols, which cool the Earth—referred to as conventional SO₄ SAI in the following.”

L109: “...suggesting a roughly linear relationship between SO₄ aerosol loads and the corresponding ERF.” I think this is more or less expected, but it would not hold if aerosol microphysics were taken into account.

Response: Thank you for the comment. We agree that the roughly linear relationship is expected under our current setup, which does not account for aerosol microphysics — including particle formation, growth, and interactions in the atmosphere. We have clarified this limitation in the updated manuscript.

“For the SO₄ deployments, the ERF exhibits little sensitivity to the vertical location of the aerosol layer, with an ERF of ~0.12 W m⁻² for 0.5 Tg of SO₄ and ~1.3 W m⁻² for 5.0 Tg of SO₄,

suggesting a roughly linear relationship between SO₄ aerosol loads and the corresponding ERF, as simulated in the models without considering aerosol microphysics.”

L152: What altitude were the aerosols injected at in these coupled simulations? I see this information in Extended Data Table 1, but it might be helpful to mention it here as well.

Response: Thank you for the suggestion. We have added the altitude of aerosol deployment in the coupled simulations to the main text for clarity in the updated manuscript.

L297: “0.5 Tg BC and 5.0 Tg SO₄ aerosols are prescribed separately at the most effective layer found in the previous series of fixed-SST simulations.” According to Extended Data Table 1, the aerosols are placed at the second-highest level. However, based on GFDL-CM2 results, these are not the levels where the ERF is maximized. Could you clarify?

Response: Thank you for the question. In our simulations, the effective radiative forcing (ERF) values at the two highest model levels (L1 and L2) are nearly identical. Specifically, in the AM2.5 model, L1 and L2 exhibit similar radiative forcing values, though L2 is slightly lower than L1. Conversely, in the AM2.1 model, L2 shows a mildly stronger forcing than L1. Given this small difference in ERF between these two levels, we opted to prescribe aerosols at L2 rather than L1. This decision was based on practical considerations—placing aerosols at a slightly lower altitude (L2) could be more feasible in real-world geoengineering scenarios, potentially reducing deployment challenges and associated costs. Thus, while L1 and L2 produce nearly the same ERF in our model, we chose L2 as a pragmatic choice rather than due to a significant difference in effectiveness.

We have incorporated this explanation into the updated manuscript as follows:

“These climate intervention simulations are initialized from pre-industrial conditions and conducted by prescribing 0.5 Tg BC and 5.0 Tg SO₄ aerosols at the second-highest level (~7.4 hPa), respectively. This level is considered the most effective deployment altitude due to its near-maximum ERF—only slightly lower than the highest value in GFDL-AM2.5 but the highest in GFDL-AM2.1. Additionally, deploying aerosols at this altitude is expected to be more practical, requiring less energy and logistical effort than reaching the highest level.”

L167-169: “Rapid precipitation decreases are found in both of these climate intervention simulations (Fig. 3b and Extended Data Fig. 4b). In particular, the fast precipitation decline in the BC simulations occurs without significant surface temperature changes.”

I’m not sure I fully follow this point. The fast precipitation response appears quite small, as the intercept in Extended Data Fig. 4b is 0.01 mm for both cases. You suggest that the response would be larger for BC, but the figure doesn’t seem to support this. That said, I was expecting to see a larger reduction in precipitation given the linear dependence between absorbed radiation and fast precipitation response (as shown in Samset et al., 2016, and Laakso et al., 2024). This relationship seems to hold for absorbing aerosols in the troposphere or lower stratosphere, based on earlier studies, but perhaps the dynamics differ when aerosols are placed at the highest levels of the model. Could you clarify this?

Samset, B. H., et al., Geophys. Res. Lett., 43, 2782–2791, <https://doi.org/10.1002/2016GL068064>, 2016

Laakso, A., et al., Earth Syst. Dynam., 15, 405–427, <https://doi.org/10.5194/esd-15-405-2024>, 2024.

Response: Thank you for the thoughtful comment and pointing to the two literatures. Our initial expectation was that BC geoengineering would lead to an increase in fast precipitation, as it counteracts the direct radiative effects of CO₂ (i.e., suppressing fast precipitation). Since elevated CO₂ typically suppresses fast precipitation by reducing outgoing longwave radiation (or increasing atmospheric absorption in the thermodynamic constraints on precipitation, $-L\Delta P = \Delta R_{TOA} - \Delta R_{SFC} + \Delta SH = \Delta R_{ATM} + \Delta SH$) and stabilizing the atmosphere (as shown in Samset et al., 2016, and Laakso et al., 2024), we hypothesized that BC-driven warming and the corresponding negative radiative forcing at the TOA would partially offset this suppression, allowing precipitation to increase.

Most previous studies on absorbing aerosols have focused on tropospheric aerosol loading — particularly in the lower troposphere — where their impacts on atmospheric stability and energy fluxes follow well-established relationships tied to positive radiative forcing (e.g., Smith et al., 2018). In contrast, our BC geoengineering scenario places aerosols at much higher altitudes, potentially shifting the thermodynamic and radiative constraints on precipitation responses.

However, as you noted, our simulations did not show the anticipated increase in fast precipitation. Instead, we observed a small reduction in the fast precipitation response (Fig. 3b, Extended Data Fig. 5b), suggesting an unknown atmospheric mechanism—likely within the troposphere. This appears to enhance negative radiative forcing at the surface through anomalous atmospheric absorption, despite the radiative forcing at the TOA and tropopause remaining identical due to radiative transfer equilibrium. The lower troposphere likely contributes to this anomalous absorption, possibly driven by unexpected increases in water vapor.

This unexpected result highlights the need for further investigation into how stratospheric aerosol deployment influences lower atmospheric processes — particularly the coupling between stratospheric temperature changes and tropospheric moisture variability. We have revised the manuscript to incorporate this explanation as follows:

“Rapid precipitation decreases are found in both of these climate intervention simulations (Fig. 3b and Extended Data Fig. 5b). In particular, the fast precipitation decline in the BC simulations occurs without significant surface temperature changes. This contrasts with our initial expectation that upper stratospheric BC deployment would increase precipitation by counteracting the direct radiative effects of CO₂ — a departure from the precipitation decreases typically associated with tropospheric BC aerosols and their atmospheric absorption (Samset et al., 2016). Typically, an increase in CO₂ suppresses fast precipitation by reducing outgoing longwave radiation (or increasing atmospheric absorption) due to energetic constraints, alongside atmospheric stabilization (e.g., Samset et al., 2016; Laakso et al., 2024). Preliminary analysis suggests that this suppression may stem from anomalous infrared absorption in the lower troposphere, possibly driven by unexpected increases in water vapor.”

Related to Extended Data Fig. 4a: The relationship between TOA energy imbalance and surface temperature appears quite scattered, whereas it typically shows a clearer linear response. However, given the relatively small response here compared to, for example, 4xCO₂ simulations, this might simply be normal noise. Do you think this is expected behavior in GFDL-CM2.1, or could it be a result of this unusual case, where model perturbations are applied at the uppermost levels?

Response: Thank you for the question. The apparent scatter in the relationship between TOA energy imbalance and surface temperature is largely due to the relatively small magnitude of the radiative forcing in this case — resulting in a narrower visual range. This contrasts with stronger

forcing scenarios, such as $4\times\text{CO}_2$ experiments, where a clearer linear relationship emerges, driven by the combination of stronger forcing, larger temperature responses, and an improved signal-to-noise ratio. To better illustrate this, we generated an identical plot but scaled to match the visual range used in a 150-year $4\times\text{CO}_2$ GFDL-CM2.1 simulation (Figure R2). When viewed on this broader scale, the relationship appears more linear, and the scatter diminishes. The scatter is even weaker than the latter period of $4\times\text{CO}_2$ simulations (related discussion will be found in the following).

Figure R2. (a) Scatter plots of flipped ensemble-mean surface air temperature anomalies versus flipped TOA energy imbalance, for the GFDL-CM2.1 model simulations forced with 0.5 Tg BC and 5 Tg SO_4 aerosols separately, by horizontal-uniformly prescribing the targeted aerosols at the second highest sigma level of aerosol inputs, beginning from the pre-industrial control runs. The individual dots represent the ensemble mean of 3 members with different conditions taken from a pre-industrial control simulation by the GFDL-CM2.1 model. The anomalies are flipped to match the conventional Gregory plots, which are designed for global warming cases. Therefore, the y-intercepts noted in the plots need to be interpreted by reversing signs. (b) The conventional Gregory plot of a 150-year $4\times\text{CO}_2$ GFDL-CM2.1 simulation. The subplot (a) is scaled to match the visual range used in a 150-year $4\times\text{CO}_2$ GFDL-CM2.1 simulation (b).

It's also worth noting that the values shown in Figure R2a and Extended Data Fig. 5a (previously Extended Data Fig. 4a) represent the ensemble mean of three simulations. This averaging reduces the influence of internal variability, making it less likely that the scatter is purely due to noise. To quantify this, we computed the slope and standard deviation for each individual ensemble member and that of their ensemble mean (Table R1), as well as for the corresponding simulations conducted with GFDL-CM2.5-FLOR. The results show that BC aerosol simulations exhibit relatively consistent behavior across ensemble members, while SO_4 simulations show more variability — suggesting that internal noise plays a larger role in the latter case.

Additionally, our comparison with GFDL-FLOR simulations reveals similar behavior to GFDL-CM2.1, suggesting that this response is not unique to one model configuration. However, given the small radiative forcing involved, it is to some extent plausible that internal variability contributes significantly to the scatter, weakening the overall signal — particularly in the SO_4 case. This raises an important consideration: whether the consistency observed in the BC ensemble mean

reflects a robust model response or, to some extent, results from a statistical coincidence where all three ensemble members aligned in the same direction.

Table R1. the slope and standard deviation for each individual ensemble member simulated by the GFDL-CM2.1 model and that of their ensemble mean, as well as for the corresponding simulations conducted with GFDL-CM2.5-FLOR.

	GFDL-CM2.1				GFDL-CM2.5-
	ENS. #1	ENS. #2	ENS. #3	ENS. mean	FLOR
0.5 Tg BC	-1.312 ± 0.177	-1.456 ± 0.200	-1.495 ± 0.216	-1.459 ± 0.146	-1.468 ± 0.201
5.0 Tg SO ₄	-1.576 ± 0.234	-1.142 ± 0.179	-1.773 ± 0.239	-1.505 ± 0.191	-1.119 ± 0.209

Fig. 4: This is a minor point, but I would suggest using a different map projection to better highlight differences at higher latitudes and near the Prime Meridian.

Response: Thank you for the comment. We have updated Fig. 4 and Extended Data Fig. 6 with a different map projection to better highlight differences at higher latitudes and near the Prime Meridian.

L227: A small detail too, but there wasn't an actual "injection" in these simulations. As mentioned in the next line, the study focused on the effects of different aerosol layer locations. Results could differ significantly if aerosol injection and transport processes were explicitly simulated.

Response: Thank you for the comment. We have replaced all instances of "injection"/"injecting" describing our simulation setup with "deployment"/"deploying" in the manuscript to more accurately reflect the experimental design.

L279-L287: How are aerosols represented in the model? Do they have an assumed fixed size? Is any water uptake considered, or are these treated as dry aerosols?

Response: Thank you for the question. In our model, aerosols are treated as externally mixed and follow a lognormal size distribution, with geometric mean radius and standard deviation values based on established studies (Haywood & Ramaswamy, 1998). For dry sulfate (SO₄), we assume a geometric mean radius of 0.05 μm and a geometric standard deviation of 2.0, consistent with previous studies (Kiehl & Briegleb, 1993; Haywood et al., 1997). Black carbon (BC) follows a geometric mean radius of 0.0118 μm with the same standard deviation (Haywood et al., 1997).

The model considers only the direct aerosol effects, with optical properties derived from Haywood & Ramaswamy (1998) and Haywood et al. (1999). Regarding water uptake, BC is treated as a dry aerosol with fixed optical properties, while sulfate aerosols' optical properties are modeled as relative-humidity-dependent. However, given the relatively low and stable stratospheric relative humidity, SO₄ optical depth remains largely unchanged in our simulations. We have incorporated this clarification of the aerosol representation into the updated manuscript.

"Aerosols are treated as externally mixed and follow a lognormal size distribution, with geometric mean radius and standard deviation values derived from established studies (Haywood & Ramaswamy, 1998). Specifically, sulfate (SO₄) aerosols are assigned a geometric mean radius of 0.05 μm with a geometric standard deviation of 2.0, consistent with Kiehl & Briegleb (1993) and Haywood et al. (1997). Black carbon (BC) aerosols have a smaller geometric mean radius of 0.0118 μm, with the same standard deviation (Haywood et al., 1997). The model includes only direct aerosol effects, with optical properties derived from Haywood & Ramaswamy (1998) and

Haywood et al. (1999). Water uptake is handled differently for each aerosol type: BC is modeled as a dry aerosol with fixed optical properties, while SO₄ aerosols' optical properties adjust based on relative humidity (RH). However, because stratospheric RH remains relatively low and stable, SO₄ optical depth shows little variability in our simulations.”

L307: I'm not entirely sure I understand the purpose of including the GFDL-AM2.1 model here. While using an additional model is certainly always valuable, did it provide any specific insights or added value beyond what was obtained from GFDL-AM2.5?

Response: Thank you for the question. The inclusion of GFDL-AM2.1 in our study serves two key purposes:

1. Consistency with ERF diagnostic recommendations: Forster et al. (2016) recommend 30-year simulations for robust effective radiative forcing (ERF) estimates with minimal noise. GFDL-AM2.1 allows us to achieve these longer runs more efficiently, making it a practical choice for obtaining clearer diagnostics.

2. Alignment with coupled simulations: Our coupled model simulations rely heavily on GFDL-CM2.1, as its lower computational cost allows for multi-ensemble simulations. Since CM2.1 shares the same atmospheric component as AM2.1, including AM2.1 ensures consistency between the atmosphere-only and coupled simulations, allowing for more direct comparisons and improving the reliability of our multi-model analysis.

We have incorporated this reasoning into the updated manuscript to clarify the value of including GFDL-AM2.1.

“As an initial assessment of the model dependence within the results presented here and to enable multiple ensemble members for reducing the impact of internal variability, we incorporate a less computationally-expensive GFDL model (GFDL-AM2.1 and corresponding coupled model GFDL-CM2.1) for additional corroboration of our results (Extended Data Table 2). This serves two key purposes: first, it aligns with established ERF diagnostic recommendations (Forster et al., 2016), which suggest 30-year simulations to minimize noise — a more practical approach with AM2.1 due to its lower computational cost. Second, since our multi-ensemble coupled simulations rely on GFDL-CM2.1, which shares the same atmospheric component, including AM2.1 ensures consistency between atmosphere-only and coupled model experiments. Notably, the aerosol representation in AM2.1 remains identical to that in AM2.5.”

Vertical levels have been referred to as sigma levels in several figures. I would prefer to use pressure or altitude levels/units as they are easier to understand.

Response: Thank you for the comment. We have replaced sigma levels with approximated pressure levels in all relevant figures for clearer interpretation, as you suggested. Additionally, we have included an explanation of this conversion process in the Materials and Methods section of the updated manuscript.

“In this study, a present-day simulation is integrated for 10 years as a control run in GFDL-AM2.5 with the prescribed SST and sea ice concentration of the monthly climatology of the period 1986-2005, along with GHG, aerosol, and ozone concentrations fixed at corresponding levels. To explore the altitude dependence in stratospheric aerosol deployments, we prescribe 0.5 Tg of horizontally-uniform aerosols separately within each of the highest 7 sigma levels of the aerosol input file for both BC and SO₄. For clearer communication to the public, we convert these

sigma levels to approximate pressure levels by multiplying by 1000 hPa and present the corresponding pressure levels in the main text.”

Reference

- Boucher, O., Randall, D., Artaxo, P., Bretherton, C., Feingold, G., Forster, P., et al. (2013). Clouds and aerosols. In T. F. Stocker, D. Qin, G.-K. Plattner, M. Tignor, S. K. Allen, J. Doschung, et al. (Eds.), *Climate change 2013: The physical science basis. Contribution of working group I to the fifth assessment report of the intergovernmental panel on climate change* (pp. 571–657). Cambridge University Press.
- Forster, P. M., Richardson, T., Maycock, A. C., Smith, C. J., Samset, B. H., Myhre, G., & Schulz, M. (2016). Recommendations for diagnosing effective radiative forcing from climate models for CMIP6. *Journal of Geophysical Research: Atmospheres*, **121**, 12,460–12,475.
- Forster, P., Storelvmo, T., Armour, K., Collins, W., Dufresne, J., Frame, D., et al. (2021). The Earth's energy budget, climate feedbacks, and climate sensitivity. In V. Masson-Delmotte, P. Zhai, A. Pirani, S. L. Connors, C. Péan, S. Berger, et al. (Eds.), *Climate change 2021: The Physical Science Basis. Contribution of Working Group I to the Sixth Assessment Report of the Intergovernmental Panel on climate change*. Cambridge University Press.
- Gao, R.-S., Rosenlof, K. H., Kärcher, B., Tilmes, S., Toon, O. B., Maloney, C., and Yu, P. (2021). Toward practical stratospheric aerosol albedo modification: Solar-powered lofting. *Science Advances*, **7**, eabe3416.
- Haywood, J. M., Roberts, D. L., Slingo, A., Edwards, J. M., & Shine, K. P. (1997). General Circulation Model calculations of the direct radiative forcing by anthropogenic sulfate and fossil-fuel soot aerosol. *Journal of Climate*, **10**(7), 1562–1577.
- Haywood, J. M., & Ramaswamy, V. (1998). Global sensitivity studies of the direct radiative forcing due to anthropogenic sulfate and black carbon aerosols. *Journal of Geophysical Research*, **103**(D6), 6043–6058.
- Haywood, J. M., Ramaswamy, V., & Soden, B. J. (1999). Tropospheric aerosol climate forcing in clear-sky satellite observations over the oceans. *Science*, **283**(5406), 1299–1303.
- Haywood, J. M., Jones, A., Johnson, B. T., & McFarlane Smith, W. (2022). Assessing the consequences of including aerosol absorption in potential stratospheric aerosol injection climate intervention strategies. *Atmospheric Chemistry and Physics*, **22**(9), 6135–6150.
- He, H., Kramer, R. J., Soden, B. J., & Jeevanjee N. (2023). State dependence of CO₂ forcing and its implications for climate sensitivity. *Science*, **382** (6674), 1051-1056.
- Jeevanjee, N., Seeley, J. T., Paynter, D., & Fueglistaler, S. (2021). An Analytical Model for Spatially Varying Clear-Sky CO₂ Forcing. *Journal of Climate*, **34**(23), 9463-9480.
- Kiehl, J. T., & Briegleb, B. P. (1993). The relative roles of sulfate aerosols and greenhouse gases in climate forcing. *Science*, **260**(5106), 311–314.
- Laakso, A., Visioni, D., Niemeier, U., Tilmes, S., & Kokkola, H. (2024) Dependency of the impacts of geoengineering on the stratospheric sulfur injection strategy – Part 2: How changes in the

hydrological cycle depend on the injection rate and model used, *Earth System Dynamics*, **15**, 405–427.

McClellan, J., Keith, D. W., & Apt, J. (2012). Cost analysis of stratospheric albedo modification delivery systems. *Environmental Research Letters*, **7**(3), 034019.

Myhre, G., Shindell, D., Bréon, F.-M., Collins, W., Fuglestvedt, J., Huang, J., et al. (2013). Anthropogenic and natural radiative forcing. In T. F. Stocker, D. Qin, G.-K. Plattner, M. Tignor, S. K. Allen, J. Doschung, et al. (Eds.), *Climate change 2013: The physical science basis. Contribution of working group I to the fifth assessment report of the intergovernmental panel on climate change* (pp. 659–740). Cambridge University Press.

National Academies of Sciences, Engineering, and Medicine. (2021). *Reflecting sunlight: Recommendations for solar geoengineering research and research governance*. The National Academies Press.

Samset, B. H., Myhre, G., Forster, P. M., Hodnebrog, Ø., Andrews, T., Faluvegi, G., et al. (2016). Fast and slow precipitation responses to individual climate forcings: A PDRMIP multimodel study. *Geophysical Research Letters*, **43**(6), 2782–2791.

Smith, C. J., Kramer, R. J., Myhre, G., Forster, P. M., Soden, B. J., Andrews, T., et al. (2018). Understanding rapid adjustments to diverse forcing agents. *Geophysical Research Letters*, **45**, 12,023–12,031.

Smith, C. J., Kramer, R. J., Myhre, G., Alterskjær, K., Collins, W., Sima, A., et al. (2020). Effective radiative forcing and adjustments in CMIP6 models. *Atmospheric Chemistry and Physics*, **20**(16), 9591–9618.

Smith, W., & Wagner, G. (2018). Stratospheric aerosol injection tactics and costs in the first 15 years of deployment. *Environmental Research Letters*, **13**(12), 124001.

Reviewer #2 (Remarks to the Author):

This manuscript presents a new approach to stratospheric aerosol injection (SAI) for climate intervention by introducing absorptive aerosols, particularly black carbon (BC), in the upper stratosphere around the 10 hPa level (near the emission level of CO₂). The study hypothesizes that this strategy could weaken CO₂'s greenhouse effect by increasing outgoing longwave radiation at the top of the atmosphere, offering potentially ~12x more efficient cooling than traditional reflective aerosols injected into the lower-mid stratosphere. Global model simulations explore the radiative effects of this method, highlighting potential benefits and calling for further research on unintended impacts. The paper aligns with the scope of Communications Earth & Environment and would likely interest the broader atmospheric and climate science community.

Response: We thank the reviewer for the thoughtful and constructive feedback that helped us further advance the manuscript. We address individual comments below.

However, I have several major concerns and questions regarding the analysis and conclusions:

1. The air density at the 10 hPa level is extremely low, which challenges the long-term suspension of aerosols, compared to traditional SAI proposals targeting the lower-mid stratosphere. How long can BC remain at these high altitudes? Assessing particle lifetimes at this altitude and considering logistical requirements would strengthen the study's feasibility assessment.

Response: Thank you for the insightful comment. In our current study, we prescribe aerosol concentrations around 10 hPa without explicitly simulating their lifetime, transport, or removal processes — bypassing the more realistic but computational-expensive approach of injection-based simulations. In other words, aerosol lifetimes were not accounted for, as the concentrations were fixed at the target altitude throughout the simulation. This setup, while useful for isolating radiative effects, inherently overlooks the complexities of aerosol persistence at such high altitudes. We acknowledge that the extremely low air density around 10 hPa presents significant challenges for sustaining aerosols over time — a key factor when evaluating the feasibility of BC aerosols for climate intervention.

Although we do not model aerosol lifetimes directly, prior studies on sulfate aerosols suggest that higher-altitude injections generally result in longer aerosol residence times due to placement within the upper branch of the Brewer-Dobson circulation (BDC) (Lee et al., 2023; Tilmes et al., 2017, 2018). This prolonged residence time arises because aerosols injected deeper into the upper BDC experience slower descent rates and reduced removal via sedimentation, leading to a higher aerosol burden per unit of injected material.

However, extended aerosol lifetime can also promote particle growth through coagulation, which reduces radiative efficiency by increasing particle size beyond the optimal effective radius for reflecting sunlight (Dykema et al., 2016). Larger aerosols not only scatter sunlight less effectively but also settle out of the stratosphere more quickly, offsetting the benefits of a longer sulfate aerosol lifetime. These competing processes — extended residence time versus particle growth — have been shown to influence sulfate geoengineering effectiveness, with outcomes varying based on altitude and injection latitude (Kleinschmitt et al., 2018; Tilmes et al., 2017).

For BC aerosols specifically, additional complexities arise. Self-lofting, driven by absorption-induced heating, may help maintain BC aerosols at high altitudes, but this same heating could also

disrupt stratospheric circulation, potentially altering aerosol lifetimes and spatial distribution (e.g., Gao et al., 2021; Haywood et al., 2022). While our study does not explicitly capture these dynamics, especially the interaction between aerosols and stratospheric circulation, we recognize that a more detailed assessment of BC lifetime—accounting for transport, sedimentation, and removal processes—would strengthen the feasibility analysis. Future work incorporating interactive aerosol physics will be essential for more accurately quantifying BC aerosol behavior under realistic atmospheric conditions.

We have revised the manuscript to include this discussion, acknowledging these limitations and emphasizing the need for further investigation into BC aerosol lifetimes and transport under more realistic atmospheric conditions.

“We acknowledge that key uncertainties remain in the transport, microphysical evolution, and sedimentation of absorbing aerosols. Our simulations prescribe aerosol concentrations uniformly at fixed altitudes, neglecting redistribution due to transport processes such as interactions with stratospheric circulation, as well as self-lofting and its seasonal variability driven by changes in incoming solar radiation. Since the BDC plays a crucial role in aerosol transport, anomalous radiative heating could further shape aerosol distribution, modulating their lifetime and climatic impacts (e.g., Lee et al., 2023; Tilmes et al., 2017, 2018). Additionally, aerosol microphysics—governing size and compositional changes through nucleation, condensation, coagulation, hygroscopic growth, and sedimentation—can significantly affect radiative properties and further influence residence time at the intended altitude (Dykema et al., 2016; Kleinschmitt et al., 2018; Tilmes et al., 2017).”

2. The stratospheric budget of aerosols over time is not clearly explained in the paper. Specifically, do the authors prescribe constant aerosol amounts (e.g., 0.5 Tg or 5 Tg) at specific altitudes throughout the model period, or do they inject aerosols only at the beginning of the model runs? It seems that the former approach is used, but if so, BC injected at 10 hPa would sediment through the lower and middle atmosphere, impacting layers far below the target altitude. This sedimentation process, which appears underrepresented in the model, could lead to unintended warming of the lower stratosphere and tropopause, water vapor feedbacks, and ozone impacts. More importantly, this could significantly affect the model’s cooling efficiency. Therefore, it is essential for the authors to explicitly describe the aerosol injection method used in their simulations and analyze the consequences of BC aerosol sedimentation.

Response: Thank you for the thoughtful comment. You are absolutely right that the stratospheric aerosol budget and transport processes should be more explicitly described.

In our simulations, we prescribe a constant aerosol amount (e.g., 0.5 Tg or 5.0 Tg) at specific altitudes throughout the model period, rather than simulating an injection event followed by transport and removal processes. This setup ensures a fixed aerosol burden, enabling us to isolate the radiative effects without the added complexity of evolving aerosol distributions. However, we acknowledge that this approach inherently excludes key processes such as sedimentation, mixing, and self-lofting, which could result in significant deviations from a fully interactive aerosol simulation. We have clarified these details in the experiment description as follows:

“In our simulations, we prescribe a constant aerosol mass burden at specific altitudes, maintaining a horizontally uniform distribution throughout the model period. Rather than

simulating an injection event followed by transport, mixing, and removal processes, the aerosol layer remains time-invariant, with concentrations fixed at designated sigma levels. This approach simplifies the setup by excluding sedimentation and other dynamic redistributions, which would occur in fully interactive aerosol models.”

We acknowledge that BC aerosols prescribed at 10 hPa are expected to gradually descend through the stratosphere, potentially impacting layers far below the initial altitude — a process not captured in our current setup. This sedimentation would likely lead to several important consequences:

1. Efficiency Reduction & Climate Impacts: The downward transport of BC into lower altitudes would likely undermine the proposed method’s cooling efficiency. Absorptive heating from BC could lead to unintended warming of the lower stratosphere and tropopause, altering temperature gradients and stratospheric stability.

2. Water Vapor Feedbacks: As BC aerosols descend, they could enhance lower-stratospheric heating, potentially increasing water vapor concentrations. Since water vapor is a potent greenhouse gas, this could counteract some of the cooling benefits of BC injection.

3. Seasonal and Latitudinal Dependence: Due to the self-lofting effect of BC (from solar absorption and heating-induced vertical motion), we anticipate that aerosol sedimentation would be more pronounced during local winter, when the absence of incoming solar radiation prevents continued lofting. This would lead to a seasonal cycle in BC distribution, with more aerosols settling downward in winter. As a result, BC aerosols would have a reduced radiative impact during this season, as they would not absorb solar radiation and induce heating in the absence of sunlight. This seasonal variation in BC at lower altitude could, to some extent, reduce potential side effects, such as excessive lower-stratospheric warming, by limiting heating when solar radiation is absent.

4. Ozone Depletion: The downward transport of BC could also influence stratospheric chemistry by modifying ozone concentrations. While we use prescribed ozone levels in our study, previous research has shown that increased lower-stratospheric heating and water vapor changes could exacerbate ozone loss, particularly in polar regions.

5. Geographical Distribution of Effects: The structure of the BDC suggests that most of the BC sedimentation would occur at higher latitudes, where subsiding air masses transport aerosols downward more efficiently. Admittedly, this could result in regional variations in stratospheric heating, with enhanced warming over high-latitude regions during summer.

We acknowledge that our simplified setup does not capture these transport and sedimentation processes, and we have discussed these limitations in the revised manuscript. A fully interactive model, which includes aerosol transport, lofting, and removal mechanisms, would be necessary to accurately quantify BC lifetime and its impacts on stratospheric dynamics, water vapor, and ozone. Future studies incorporating such interactive dynamics will be critical for assessing the feasibility of BC-based climate intervention strategies.

“As BC aerosols move downward, they may warm the lower stratosphere, reducing the cooling efficiency of the intervention. This warming could also alter temperature gradients, destabilizing the stratosphere and potentially affecting tropospheric circulation patterns. Moreover, the enhanced lower-stratospheric heating may increase water vapor concentrations, which could counteract the intended cooling effects by amplifying the greenhouse effect and exacerbating ozone depletion.”

3. Given the likely path of BC aerosols through the Brewer-Dobson circulation, much of the injected BC may eventually sediment at the surface in high-altitude or polar regions. This could significantly alter the surface albedo of ice and snow, potentially offsetting the cooling effects and causing a positive feedback. This limitation should also be discussed.

Response: Thank you for the thoughtful comment. We agree that BC deposition onto snow and ice presents a significant limitation. When BC aerosols settle on these surfaces, they reduce albedo, increasing solar absorption and accelerating melting — particularly in high-latitude and high-altitude regions, where persistent snow and ice play a crucial role in the climate system. This effect can lead to increased surface temperatures and trigger feedback mechanisms that may offset the intended cooling. Given the expected transport of BC through the Brewer-Dobson circulation, some fraction of the aerosols may eventually reach these regions, with deposition likely more pronounced during winter due to seasonal variations in self-lofting and sedimentation. While wintertime deposition has minimal immediate radiative impact due to limited sunlight, the warming effect would emerge once sunlight returns, potentially enhancing melting and reinforcing positive feedbacks. Although our current simulations do not explicitly track BC deposition, we acknowledge this limitation and have added discussion on its potential to reduce the intervention’s overall climate effectiveness, as follows:

“In addition, when BC aerosols sediment onto snow and ice surfaces, they reduce surface albedo, causing the snow and ice to absorb more solar radiation rather than reflecting it. This effect accelerates melting, increases surface temperatures, and triggers feedback mechanisms that could offset some of the cooling effects of BC aerosols.”

4. The warming of the upper stratosphere induced by BC aerosols could alter stratospheric dynamics, such as changes in the Brewer-Dobson circulation and polar vortex behavior. Additionally, BC interactions with stratospheric ozone could exacerbate ozone depletion, leading to higher surface UV radiation and potentially counteracting any cooling effects. Moreover, warming in the upper stratosphere could increase water vapor, especially if tropopause temperatures rise. These effects could diminish or offset the cooling by adding additional greenhouse gases. Were these potential side effects considered in the model simulations?

Response: We appreciate the reviewer’s insightful comments regarding the potential dynamical and chemical side effects of upper stratospheric BC heating. These are indeed important considerations, though they are not fully resolved in our current simulations.

Stratospheric Dynamics: Brewer-Dobson Circulation and Polar Vortex

Our results suggest that BC-induced warming in the upper stratosphere could alter stratospheric dynamics, particularly the Brewer-Dobson circulation (BDC) and polar vortex behavior. The BDC transports air from the tropics to higher latitudes in the stratosphere, with descending motion at mid-to-high latitudes. In contrast to the acceleration observed for sulfate aerosols, our simulations indicate a slowdown of the BDC in response to BC heating (Figure R3).

Figure R3. Residual mass stream function anomalies (shading) in response to in response to horizontal-uniformly prescribed 0.5 Tg BC (**top**) and 5.0 Tg SO₄ (**bottom**) aerosols at each of the highest 7 sigma levels of aerosol inputs. From **left to right**, panels show results for aerosols placed progressively lower in the stratosphere. Contours in each subplot display the climatological distribution from the control simulation. The residual circulation is calculated following Andrews et al. (1987) and Salby (1996).

Since most of the stratospheric temperature response is driven by radiative heating, we do not see a significant change in the annual mean temperature gradient. However, in winter, when BC aerosols experience less solar absorption at high latitudes, a stronger equator-to-pole temperature gradient could develop, potentially strengthening the stratospheric polar vortex. This, in turn, may influence stratosphere-troposphere coupling. Previous studies suggest that an intensified polar vortex can shift the North Atlantic Oscillation (NAO) toward a more positive phase, strengthening the Atlantic jet stream and altering storm tracks, particularly over Europe and North America (Baldwin et al., 2003; Hurrell, 1995). While our study does not focus on these tropospheric impacts, we acknowledge that such circulation changes could arise from the seasonal evolution of stratospheric temperature gradients.

“Beyond their direct radiative impacts, BC and SO₄ aerosols also influence stratospheric dynamics (e.g., Bednarz et al. 2023a, 2023b), warranting further consideration. Our preliminary results suggest that BC-induced warming in the upper stratosphere can modify both the Brewer-Dobson circulation (BDC) and the polar vortex. Unlike sulfate aerosols, which accelerate the BDC by enhancing tropical upwelling, BC-induced heating appears to slow it down, potentially altering stratospheric ozone transport and distribution (Extended Data Fig. 8). Seasonal variations in BC radiative heating may also strengthen the equator-to-pole temperature gradient in winter, reinforcing the polar vortex. This has implications for stratosphere-troposphere coupling, as a stronger polar vortex has been linked to a positive phase of the North Atlantic Oscillation, which in turn influences storm tracks, temperature anomalies, and precipitation patterns across midlatitude regions, particularly over Europe and North America (Baldwin et al., 2003; Hurrell, 1995). However, our model lacks interactive stratospheric chemistry and has a relatively low model top, limiting its ability to capture the full extent of these interactions, particularly given that the available monthly wind field outputs adopted to calculate residual circulation (Andrews et al., 1987; Salby, 1996) smooth out short-term fluctuations and lose fine-scale temporal details. Nonetheless, our findings suggest that BC-induced upper stratospheric warming may influence large-scale circulation patterns in ways distinct from SO₄ aerosol geoengineering.”

Stratospheric Chemistry: Ozone and Water Vapor Feedbacks

Ozone depletion is another critical factor that could amplify some of these effects. Since our simulations prescribe ozone levels rather than simulating interactive stratospheric chemistry, we do not explicitly resolve BC-driven ozone depletion. However, it is well established that warming in the upper stratosphere can enhance catalytic ozone destruction, particularly in the presence of halogens (Solomon et al., 1996, 1998). Additionally, a BDC slowdown could reduce ozone transport from the tropics to higher latitudes, further exacerbating ozone depletion in some regions. Reduced ozone levels would increase ultraviolet (UV) radiation through the atmospheric layer, which could further reduce ozone concentrations in the lower stratosphere due to enhanced photodissociation. Additionally, BC aerosols themselves absorb UV radiation (Ramanathan & Carmichael, 2008), which may locally limit UV availability for ozone breakdown, introducing further complexity to the net effect on ozone and radiation.

Water vapor feedbacks could further influence the radiative impacts of BC. Our analysis shows that BC heating in the upper stratosphere can increase stratospheric water vapor concentrations, comparable to the moistening observed when sulfate aerosols are placed at ~100 hPa. Since water vapor is a potent greenhouse gas, this increase could offset some of the net cooling effect of BC injection by enhancing longwave radiation trapping in the stratosphere (Forster & Shine, 2002). Additionally, increased stratospheric water vapor could contribute to ozone depletion through heterogeneous chemistry (Solomon et al., 1986).

“A concern of both intervention strategies is the warming of tropopause temperatures, which can trigger a cascade of negative consequences — including increased water vapor input into the stratosphere, reductions in stratospheric ozone concentrations (Mills et al., 2008; Kravitz et al., 2012), and disruptions to both stratospheric and tropospheric circulations (Bednarz et al. 2023a, 2023b; Hueholt et al., 2023; Simpson et al., 2019; Visioni et al., 2020). As expected, the maximum atmospheric warming closely coincides with the location of the aerosol layer for both SO₄ and BC, although BC aerosols generate significantly more warming due to their greater absorptivity (Extended Data Fig. 7a). The higher and thinner the atmosphere, the greater the warming. It is important to note that higher temperatures may accelerate the catalytic reaction rates of ozone-depleting processes (Solomon et al., 1996, 1998), potentially resulting in greater ozone loss, although these effects are not considered in our simulations, which prescribe ozone concentrations. Aerosols in the lower stratosphere induce more warming around tropopause and upper troposphere, weakening the “freeze-drying” process and consequently increasing the stratospheric water vapor mixing ratio (Extended Data Fig. 7b; Mote et al., 1996; Randel et al., 2006). This water vapor increase contributes additional radiative forcing (Forster & Shine, 2002) and, more importantly, may catalyze ozone depletion through the formation of hydroxyl (OH) radicals (Solomon et al., 1986).”

“On the other hand, BC aerosols also absorb ultraviolet (UV) radiation (Ramanathan & Carmichael, 2008), which could locally reduce UV flux and mitigate ozone depletion by limiting ozone photodissociation, particularly for ozone below the aerosol layer. However, our model does not include interactive ozone chemistry, constraining our ability to fully quantify these effects.”

Future Considerations

While these effects were not fully resolved in our current study, we recognize their importance and have explicitly discussed them in the revised manuscript. Future work should incorporate

interactive stratospheric chemistry, dynamic ozone feedbacks, and a fully resolved stratosphere-troposphere coupling to better assess these complex interactions.

“Future studies incorporating interactive aerosol dynamics, stratospheric chemistry, and higher model tops are essential for refining these findings and improving our understanding of BC-induced stratospheric changes.”

5. Simulations use a pre-industrial baseline, which, while simplifying the analysis, does not reflect modern greenhouse gas concentrations, which are the primary motivation for the study. Given that weakening CO₂ forcing is a central aim, it would be more informative to model the intervention under current or projected CO₂ levels to provide a realistic view of its effectiveness in today’s atmospheric condition.

Response: Thank you for the thoughtful comment. We acknowledge the importance of evaluating BC interventions under present or projected greenhouse gas concentrations to ensure a more policy-relevant assessment. While our primary analysis uses a pre-industrial baseline for simplicity and clearer attribution of aerosol-induced changes, we have also conducted simulations under increased CO₂ levels. Specifically, we utilize results from two doubling-CO₂ (2×CO₂) scenarios:

1. Transient Forcing Scenario (1pct2×CO₂, non-equilibrium): CO₂ increases by 1% per year until doubling (~year 70), after which it is held constant. The climate interventions are applied from year 100 of 1pct2×CO₂, when the system has partially equilibrated to the doubled CO₂ forcing.

2. Equilibrium Doubling Scenario (2×CO₂, equilibrium): CO₂ concentrations are abruptly doubled and held constant throughout the simulation. The simulation runs until the climate reaches equilibrium, at which point targeted aerosols are introduced.

In both scenarios, upper stratospheric BC deployment produces stronger cooling than sulfate aerosols (Figures R4 & R5). However, contrary to our initial expectation, the cooling effectiveness of BC aerosols does not scale linearly with increasing CO₂ forcing. This could be due to state-dependent climate feedbacks, where a warmer base state — accompanied by a wetter atmosphere — dampens further cooling, limiting the incremental cooling response. Additionally, tropospheric base-state dependence in the BC aerosol forcing may play a role, as changes in cloud cover, water vapor distribution, and stratosphere-troposphere coupling could modify the overall radiative impact.

Figure R4. Global mean (a) surface air temperature and (b) precipitation anomalies in climate intervention simulations starting from a non-equilibrium CO₂ doubling case simulated by GFDL-CM2.1. The non-equilibrium CO₂ doubling state refers to year 100 of the 1pct2×CO₂ simulations, where CO₂ concentrations increase by 1% per year until doubling at year 70, then remain fixed for the remainder of the simulation. Anomalies are calculated relative to the climatological mean of a 30-year period centered on year 100 (years 85–115) of the 1pct2×CO₂ simulations. The solid line represents the ensemble mean of 3 members from the GFDL-CM2.1 model initiated from year 100 of 1pct2×CO₂ members. The corresponding light shading represents the range from minimum to maximum for every year.

Figure R5. Global mean (a) surface air temperature and (b) precipitation anomalies in climate intervention simulations starting from an equilibrium CO₂ doubling case simulated by GFDL-CM2.1. The equilibrium CO₂ doubling state refers to the later stages of long-run, abrupt CO₂ doubling simulations, where the atmosphere has reached equilibrium with no further warming or energy imbalance. Anomalies are calculated relative to the climatological mean of the equilibrium simulations. The solid line represents the ensemble mean of 3 members from the GFDL-CM2.1 model with different initial conditions taken from the equilibrium simulations. The corresponding light shading represents the range from minimum to maximum for every year.

Further investigation is needed to disentangle these effects and understand how stratospheric BC cooling interacts with different climate states. Exploring these feedbacks and higher-end CO₂ scenarios (e.g., climate change scenarios of projected socioeconomic global changes) would improve our understanding of BC cooling under extreme warming conditions. For now, we present these results briefly to maintain the manuscript's focus.

“The effectiveness in mediating warming is observed in both a single realization of the GFDL-CM2.5-FLOR (dotted line) and the 3-member ensemble mean of the GFDL-CM2.1 (solid line and shading), as well as in the 3-member ensemble means of intervention simulations initialized from both non-equilibrium and equilibrium CO₂ doubling cases using GFDL-CM2.1 (shown in Supplementary Figures 3 & 4, though not discussed further to maintain a streamlined narrative).”

I also have some specific comments:

1. Line 54-55: Please provide relevant references for “analytic and radiative transfer calculations demonstrate the feasibility of weakening the CO₂ greenhouse effect.”

Response: Thank you for the comment. We have included the relevant references to support the statement regarding the feasibility of weakening the CO₂ greenhouse effect through analytic and radiative transfer calculations in the updated manuscript.

2. Line 114: Please explain what “sigma level” means and why this coordinate system is beneficial over pressure levels for a more general audience.

Response: Thank you for your comment. “Sigma level” refers to a vertical coordinate system used in atmospheric models, where altitude is defined relative to the local surface pressure. Specifically, sigma (σ) is expressed as the ratio of local pressure to surface pressure, with $\sigma = 1$ representing the surface and $\sigma = 0$ representing the top of the atmosphere. This system is beneficial because it better follows the curvature of the Earth and can more accurately represent atmospheric layers near the surface, which can vary significantly in pressure.

As suggested by both you and Reviewer #1, we have updated the manuscript to use “pressure levels” instead of “sigma levels” for clarity and accessibility to a more general audience.

3. Figure 1 and Figure 4: Please add units to the color bars.

Response: Thank you for your comment. We have added the appropriate units to the color bars in all color-shading figures for clarity.

Reference

Andrews, D. G., Holton, J. R., & Leovy, C. B. (1987). Middle atmosphere dynamics. Academic Press.

Baldwin, M. P., Stephenson, D. B., Thompson, D. W., Dunkerton, T. J., Charlton, A. J., & O’Neill, A. (2003). Stratospheric memory and skill of extended-range weather forecasts. *Science*, **301**(5633), 636–640.

Dykema, J. A., Keith, D. W., & Keutsch, F. N. (2016). Improved aerosol radiative properties as a foundation for solar geoengineering risk assessment. *Geophysical Research Letters*, **43**(14), 7758–7766.

Forster, P. M., & Shine, K. P. (2002). Assessing the climate impact of trends in stratospheric water vapor. *Geophysical Research Letters*, **29**(6), 10–1104.

Gao, R.-S., Rosenlof, K. H., Kärcher, B., Tilmes, S., Toon, O. B., Maloney, C., and Yu, P. (2021). Toward practical stratospheric aerosol albedo modification: Solar-powered lofting. *Science Advances*, **7**, eabe3416.

Haywood, J. M., Jones, A., Johnson, B. T., & McFarlane Smith, W. (2022). Assessing the consequences of including aerosol absorption in potential stratospheric aerosol injection climate intervention strategies. *Atmospheric Chemistry and Physics*, **22**(9), 6135–6150.

- Hurrell, J. W. (1995). Decadal trends in the North Atlantic Oscillation: Regional temperatures and precipitation. *Science*, **269**(5224), 676–679.
- Kleinschmitt, C., Boucher, O., & Platt, U. (2018). Sensitivity of the radiative forcing by stratospheric sulfur geoengineering to the amount and strategy of the SO₂ injection studied with the LMDZ-S3A model. *Atmospheric Chemistry and Physics*, **18**(4), 2769–2786.
- Lee, W. R., Visoni, D., Bednarz, E. M., MacMartin, D. G., Kravitz, B., & Tilmes, S. (2023). Quantifying the efficiency of stratospheric aerosol geoengineering at different altitudes. *Geophysical Research Letters*, **50**, e2023GL104417.
- Ramanathan, V., & Carmichael, G. (2008). Global and regional climate changes due to black carbon. *Nature Geoscience*, **1**(4), 221–227.
- Salby, M. L. (1996). *Fundamentals of atmospheric physics*. Academic Press.
- Solomon, S., Garcia, R. R., Rowland, F. S., & Wuebbles, D. J. (1986). On the depletion of Antarctic ozone. *Nature*, **321**(6072), 755–758.
- Solomon, S., Portmann, R. W., Garcia, R. R., Thomason, L. W., Poole, L. R., & McCormick, M. P. (1996). The role of aerosol variations in anthropogenic ozone depletion at northern midlatitudes. *Journal of Geophysical Research*, **101**(D3), 6713–6727.
- Solomon, S., Portmann, R. W., Garcia, R. R., Randel, W., Wu, F., Nagatani, R., et al. (1998). Ozone depletion at mid-latitudes: Coupling of volcanic aerosols and temperature variability to anthropogenic chlorine. *Geophysical Research Letters*, **25**(11), 1871–1874.
- Tilmes, S., Richter, J. H., Mills, M. J., Kravitz, B., MacMartin, D. G., Vitt, F., et al. (2017). Sensitivity of aerosol distribution and climate response to stratospheric SO₂ injection locations. *Journal of Geophysical Research: Atmospheres*, **122**(23), 12591–12615.
- Tilmes, S., Richter, J. H., Mills, M. J., Kravitz, B., MacMartin, D. G., Garcia, R. R., et al. (2018). Effects of different stratospheric SO₂ Injection altitudes on stratospheric chemistry and dynamics. *Journal of Geophysical Research: Atmospheres*, **123**(9), 4654–4673.

Reviewer #3 (Remarks to the Author):

In this paper, the authors are trying to show that the addition of black carbon in the mid-upper stratosphere (10 hPa) has a much stronger radiative effect than sulfate, which is normally used for stratospheric aerosol injection (SAI) simulations: they explain that this difference is driven by a strong LW effect by the absorbing aerosols in the case of BC.

While in general the manuscript is well written and the results interesting, there is a major flaw at this point that prevents me from understanding the results fully, and that the authors will need to address before moving further.

Response: We thank the reviewer for the thoughtful and constructive feedback, which has helped us further improve the manuscript. Below, we provide detailed responses to each individual comment.

Essentially, the description of the methods is not in depth enough to allow me to understand what exactly has been done in the simulations. The authors often use the word "prescribed" when referring to the aerosol layer. First, I'd like more details about this. Did you prescribe an optical depth uniformly? Can one see it? How does it compare to other distributions used in other papers?

Response: Thank you for the question. Yes, the aerosol layer is prescribed, meaning it remains time-invariant throughout the simulations. Specifically, the total aerosol mass is uniformly distributed globally — divided by the Earth's surface area — resulting in a consistent concentration everywhere (Figure R6).

Figure R6. Aerosol column mass density of horizontally-uniformly prescribed (a) 0.5 Tg BC aerosols and (b) 5.0 Tg SO_4 aerosols separately at each of the highest 7 sigma levels of aerosol inputs, respectively, simulated by the GFDL-AM2.1 model. Results are shown only from GFDL-AM2.1, as these variables were not saved for GFDL-AM2.5. Sigma levels are converted to approximate pressure levels by multiplying by 1000 hPa and presented in the figure. Filled and open bars represent aerosol column mass density from 0.5 Tg BC and 5.0 Tg SO_4 aerosols, respectively.

In general, the optical properties of the aerosols are also static. Black carbon (BC) aerosols, simulated as dry aerosols, maintain a static absorptive optical depth. Sulfate (SO₄) aerosols, on the other hand, have optical properties influenced by relative humidity. While stratospheric humidity changes minimally, tropospheric variations can slightly affect sulfate scattering optical depth, particularly in the Northern Hemisphere. However, this effect remains small, and global-mean optical depth anomalies capture the main response accurately (Figure R7), given the minimal latitude-dependent differences.

Figure R7. Aerosol column extinction Optical Depth (AOD) of horizontal-uniformly prescribed (a) 0.5 Tg BC aerosols and (b) 5.0 Tg SO₄ aerosols separately at each of the highest 7 sigma levels of aerosol inputs, respectively, simulated by the GFDL-AM2.1 model. Results are shown only from GFDL-AM2.1, as AOD data were not saved for GFDL-AM2.5. Sigma levels are converted to approximate pressure levels by multiplying by 1000 hPa and presented in the figure. Filled and open bars represent AOD from 0.5 Tg BC and 5.0 Tg SO₄ aerosols, respectively.

We recognize that this setup is deliberately simplified — it excludes aerosol transport, microphysics, and variable lifetimes. This approach prioritizes clearer attribution of climate responses to aerosol forcing but contrasts with emission-based models that simulate injection, dispersion, and chemical evolution. As a result, direct comparisons with such studies are less straightforward. Our method provides a clean baseline for understanding first-order climate responses, though it likely underestimates regional heterogeneity and dynamic feedbacks seen in more interactive simulations. We emphasize this as an initial exploration, intended to motivate further studies that incorporate more complex aerosol behavior and climate interactions.

We’ve revised the manuscript to clarify the simulation setup in the Materials and Methods section as follows:

“In our simulations, we prescribe a constant aerosol mass burden at specific altitudes, maintaining a horizontally uniform distribution throughout the model period. Rather than simulating an injection event followed by transport, mixing, and removal processes, the aerosol layer remains time-invariant, with concentrations fixed at designated sigma levels. This approach simplifies the setup by excluding sedimentation and other dynamic redistributions, which would occur in fully interactive aerosol models.”

Secondly, I think -but I can only guess, not having seen the actual details- that, if the authors have prescribed the aerosols, they are missing out on the potential very, very relevant effect of aerosol lifetime in their simulations. This is important because they name their simulations based on mass (0.5 or 5 Tg), and then claim that "that the reduction in global temperatures induced through this process is an order of magnitude larger (per unit aerosol mass) than the injection of more traditional reflective aerosols". But you can't make a unit mass comparison without including considerations about lifetime, and we know that, for SO₂ injections, lifetime matters greatly!

Response: Thank you for the insightful comment. You're absolutely right — aerosol lifetime is a key factor when comparing interventions by unit mass, especially for SO₂ injections. In our simulations, we prescribe a fixed aerosol burden rather than simulate injection, transport, and removal processes. This setup prioritizes isolating the climate response to a controlled, constant aerosol mass but inevitably excludes lifetime considerations.

We acknowledge this as a significant limitation, and future studies using emission-based or interactive aerosol models will be essential to fully capture the role of aerosol lifetime in determining climate impacts. To reflect this, we've expanded our discussion of the setup's constraints and the potential effects of aerosol transport. Additionally, we've updated the abstract to clarify that our comparison is based on prescribed aerosol burdens rather than injection scenarios, as suggested by both you and Reviewer #1.

"Climate model simulations indicate that the reduction in global temperatures induced through this process is an order of magnitude larger (per unit aerosol mass) than deploying more traditional reflective aerosols."

"We acknowledge that key uncertainties remain in the transport, microphysical evolution, and sedimentation of absorbing aerosols. Our simulations prescribe aerosol concentrations uniformly at fixed altitudes, neglecting redistribution due to transport processes such as interactions with stratospheric circulation, as well as self-lofting and its seasonal variability driven by changes in incoming solar radiation. Since the BDC plays a crucial role in aerosol transport, anomalous radiative heating could further shape aerosol distribution, modulating their lifetime and climatic impacts (e.g., Lee et al., 2023; Tilmes et al., 2017, 2018). Additionally, aerosol microphysics—governing size and compositional changes through nucleation, condensation, coagulation, hygroscopic growth, and sedimentation—can significantly affect radiative properties and further influence residence time at the intended altitude. As BC aerosols move downward, they may warm the lower stratosphere, reducing the cooling efficiency of the intervention. This warming could also alter temperature gradients, destabilizing the stratosphere and potentially affecting tropospheric circulation patterns. Moreover, the enhanced lower-stratospheric heating may increase water vapor concentrations, which could counteract the intended cooling effects by amplifying the greenhouse effect and exacerbating ozone depletion. In addition, when BC aerosols sediment onto snow and ice surfaces, they reduce surface albedo, causing the snow and ice to absorb more solar radiation rather than reflecting it. This effect accelerates melting, increases surface temperatures, and triggers feedback mechanisms that could offset some of the cooling effects of BC aerosols.

On the other hand, BC aerosols also absorb ultraviolet (UV) radiation (Ramanathan & Carmichael, 2008), which could locally reduce UV flux and mitigate ozone depletion by limiting ozone photodissociation, particularly for ozone below the aerosol layer. However, our model does

not include interactive ozone chemistry, constraining our ability to fully quantify these effects. Future studies incorporating interactive aerosol dynamics, stratospheric chemistry, and higher model tops are essential for refining these findings and improving our understanding of BC-induced stratospheric changes.”

To gain further insights from our current simulations, we analyzed the Brewer-Dobson Circulation (BDC) using the Transformed Eulerian Mean residual circulation approach (Andrews et al., 1987; Salby, 1996), examining potential lifetime differences between BC and sulfate aerosols driven by their interactions with large-scale circulation. Our results indicate that BC aerosols slow stratospheric circulation, while SO₄ aerosols mildly strengthen it (Figure R3). We focused on the upper branch of the BDC, aligning with our target of higher-altitude deployment. The magnitude of these circulation changes depends on aerosol altitude. Notably, the BC-induced BDC slowdown suggests that BC aerosols may persist longer in the stratosphere before settling. This, combined with the self-lofting effect of absorbing aerosols (Gao et al., 2021; Haywood et al., 2022), supports the possibility of an extended BC aerosol lifetime — though confirming this requires interactive modeling.

We've updated the manuscript accordingly as follows:

“Beyond their direct radiative impacts, BC and SO₄ aerosols also influence stratospheric dynamics (e.g., Bednarz et al. 2023a, 2023b), warranting further consideration. Our preliminary results suggest that BC-induced warming in the upper stratosphere can modify both the Brewer-Dobson circulation (BDC) and the polar vortex. Unlike sulfate aerosols, which accelerate the BDC by enhancing tropical upwelling, BC-induced heating appears to slow it down, potentially altering stratospheric ozone transport and distribution (Extended Data Fig. 8). Seasonal variations in BC radiative heating may also strengthen the equator-to-pole temperature gradient in winter, reinforcing the polar vortex. This has implications for stratosphere-troposphere coupling, as a stronger polar vortex has been linked to a positive phase of the North Atlantic Oscillation, which in turn influences storm tracks, temperature anomalies, and precipitation patterns across midlatitude regions, particularly over Europe and North America (Baldwin et al., 2003; Hurrell, 1995). However, our model lacks interactive stratospheric chemistry and has a relatively low model top, limiting its ability to capture the full extent of these interactions, particularly given that the available monthly wind field outputs adopted to calculate residual circulation (Andrews et al., 1987; Salby, 1996) smooth out short-term fluctuations and lose fine-scale temporal details. Nonetheless, our findings suggest that BC-induced upper stratospheric warming may influence large-scale circulation patterns in ways distinct from SO₄ aerosol geoengineering.”

See for instance Lee et al. (2023). Therefore, the authors should re-frame their comparison in terms of optical depth, not mass, as a prescribed mass doesn't say much about the amount of material that would need to be injected in order to achieve that layer (and that is the mass that matters!). Again, I think I would be able to have more meaningful suggestions once the details have been cleared out by the authors. These details must also include information about the size distribution of the injected aerosols, which is very relevant information. What is the effective radius for both aerosol populations?

Lee, W. R., Visioni, D., Bednarz, E. M., MacMartin, D. G., Kravitz, B., & Tilmes, S. (2023). Quantifying the efficiency of stratospheric aerosol geoengineering at different altitudes. *Geophysical Research Letters*, 50, e2023GL104417. <https://doi.org/10.1029/2023GL104417>

Response: Thank you for the comment. We acknowledge that optical depth provides a more meaningful basis for comparison than mass, particularly when evaluating shortwave radiative forcing below the aerosol layer (e.g., at the tropopause or surface) — especially in the case of absorbing aerosols like BC. Since prescribed mass alone doesn't directly translate to the amount of material required for injection, optical depth serves as a more practical and interpretable metric, particularly for the scattering aerosol deployment.

While our primary focus remains on radiative forcing at the top of atmosphere (TOA) — which offers a more comprehensive representation of the overall climate impact of a forcing agent (e.g., Boucher et al., 2013; Forster et al., 2016, 2021; Myhre et al., 2013; Smith et al., 2018, 2020) — we have incorporated an analysis of aerosol optical depth to support clearer comparisons for audiences preferring a tropopause or surface radiative forcing perspective.

Our results show that BC exhibits approximately three times the aerosol column extinction optical depth per unit mass compared to SO₄ (Figure R7), primarily due to its strong absorption despite slightly weaker scattering properties (Figure R8). Additionally, the optical depth of both BC and SO₄ aerosols shows minimal sensitivity to the vertical placement of the aerosol layer. This higher optical depth per unit mass for BC, driven by its absorption properties, is counterbalanced by its significant longwave effect. As a result, BC produces roughly three times more radiative forcing per unit optical depth than SO₄ — with this forcing arising entirely from longwave radiation, without contributing to shortwave cooling at the TOA.

Figure R8. Same as Figure R7, except showing the scattering component of aerosol column optical depth. We have expanded the discussion on aerosol optical depth to reflect these findings as follows:

“Considering that aerosol optical depth is a useful metric for assessing the negative shortwave radiative forcing, particularly below the aerosol layer (e.g., at the tropopause or surface in climate intervention simulations), especially in the presence of absorption aerosols, we diagnose

the aerosol optical depth here. This is despite our focus on radiative forcing at the TOA, as it provides a more comprehensive representation of the overall climate impact of a forcing agent (e.g., Boucher et al., 2013; Forster et al., 2016, 2021; Myhre et al., 2013; Smith et al., 2018, 2020). BC exhibits approximately three times the aerosol column extinction optical depth per unit mass compared to SO₄ (Extended Data Fig. 3), primarily due to its additional absorption, despite having slightly weaker scattering (Supplementary Figure 2). Notably, the optical depth of both BC and SO₄ aerosols shows little sensitivity to the vertical placement of the aerosol layer. This greater optical depth per unit mass of BC aerosols, particularly the absorption component, counterbalanced by their pronounced longwave effect, results in BC producing roughly three times more radiative forcing per unit optical depth than SO₄ — with this forcing arising exclusively from longwave radiation, without contributing to shortwave cooling at the TOA.”

All aerosols are treated as externally mixed, following a lognormal size distribution with geometric mean radius and standard deviation values based on established literature (Haywood & Ramaswamy, 1998). Specifically, sulfate (SO₄) aerosols are assigned a geometric mean radius of 0.05 μm with a geometric standard deviation of 2.0, consistent with Kiehl & Briegleb (1993) and Haywood et al. (1997). Black carbon (BC) aerosols are modeled with a smaller geometric mean radius of 0.0118 μm and the same standard deviation, following values from Haywood et al. (1997).

The model accounts only for direct aerosol effects, with optical properties derived from Haywood & Ramaswamy (1998) and Haywood et al. (1999). Water uptake is treated differently for each aerosol type: BC is modeled as a dry aerosol with fixed optical properties, while SO₄ aerosols' optical properties adjust based on relative humidity (RH). However, since stratospheric RH remains relatively low and stable, SO₄ optical depth shows little variability in our simulations.

Additionally, we have clarified the aerosol size distribution details in the manuscript:

“Aerosols are treated as externally mixed and follow a lognormal size distribution, with geometric mean radius and standard deviation values derived from established studies (Haywood & Ramaswamy, 1998). Specifically, sulfate (SO₄) aerosols are assigned a geometric mean radius of 0.05 μm with a geometric standard deviation of 2.0, consistent with Kiehl & Briegleb (1993) and Haywood et al. (1997). Black carbon (BC) aerosols have a smaller geometric mean radius of 0.0118 μm, with the same standard deviation (Haywood et al., 1997). The model includes only direct aerosol effects, with optical properties derived from Haywood & Ramaswamy (1998) and Haywood et al. (1999). Water uptake is handled differently for each aerosol type: BC is modeled as a dry aerosol with fixed optical properties, while SO₄ aerosols' optical properties adjust based on relative humidity (RH). However, because stratospheric RH remains relatively low and stable, SO₄ optical depth shows little variability in our simulations.”

It is also important that the authors don't simply dismiss the fact that the BC simulations have a upper stratospheric warming of 30-50 K (!!). In the conclusions, this, coupled with the acknowledgment that their models lack proper stratospheric dynamics (due to a low top I assume) and stratospheric chemistry, should really dampen the certainty the authors use also in the abstract about how effective BC would be. I assume they use prescribed ozone levels in the stratosphere, and it would be useful to discuss how a potentially very high ozone depletion would affect this, and how stratospheric circulation would change. I look forward to see these details, and I hold on judgment about how publishable these results are until then.

Response: Thank you for the thoughtful and detailed feedback. You are absolutely right — the extreme upper stratospheric warming (30–50 K) seen in our BC simulations is a critical outcome that warrants deeper discussion.

We acknowledge that our model setup — which includes prescribed aerosols, the absence of interactive stratospheric chemistry, and a relatively low model top — limits its ability to capture key feedbacks in stratospheric dynamics, particularly those involving chemistry. We have made these limitations more explicit in the manuscript.

Ozone and Water Vapor Effects

Our model uses prescribed stratospheric ozone concentrations, and we recognize that this prevents us from capturing potential ozone depletion and its associated impacts on radiative forcing and circulation. We now discuss how upper stratospheric warming may accelerate ozone-depleting catalytic reactions (Solomon et al., 1996, 1998), potentially leading to greater ozone loss, although this is not directly simulated. Additionally, BC-induced warming weakens the “freeze-drying” process near the tropopause (Mote et al., 1996; Randel et al., 2006), increasing stratospheric water vapor. This, in turn, contributes further radiative forcing (Forster & Shine, 2002) and may enhance ozone depletion via hydroxyl (OH) radical formation (Solomon et al., 1986).

Circulation Impacts

We investigated the effects of BC-induced warming on stratospheric dynamics. Unlike sulfate aerosols, which accelerate the Brewer-Dobson circulation (BDC) via enhanced tropical upwelling, BC-induced upper stratospheric heating appears to slow the BDC (Figure R3). This could potentially alter stratospheric ozone transport and distribution.

Additionally, seasonal variations in BC heating may strengthen the equator-to-pole temperature gradient during winter, reinforcing the polar vortex. This has implications for stratosphere-troposphere coupling, as a stronger polar vortex is associated with a positive phase of the North Atlantic Oscillation (NAO), influencing storm tracks, temperature anomalies, and precipitation patterns across midlatitude regions, particularly Europe and North America (Baldwin et al., 2003; Hurrell, 1995).

Concluding Acknowledgment

While our results provide valuable insights into BC’s radiative effects and dynamic impacts, we now more explicitly recognize that the absence of interactive stratospheric chemistry and the relatively low model top limit the reliability of these projections. These factors are essential for fully capturing the complex interplay between aerosol-induced warming, ozone depletion, and circulation changes. We acknowledge these limitations and have reflected them more clearly throughout the manuscript to ensure a balanced and cautious interpretation of BC’s potential as a climate intervention strategy.

“A concern of both intervention strategies is the warming of tropopause temperatures, which can trigger a cascade of negative consequences — including increased water vapor input into the stratosphere, reductions in stratospheric ozone concentrations (Mills et al., 2008; Kravitz et al., 2012), and disruptions to both stratospheric and tropospheric circulations (Bednarz et al. 2023a, 2023b; Hueholt et al., 2023; Simpson et al., 2019; Vioni et al., 2020). As expected, the maximum atmospheric warming closely coincides with the location of the aerosol layer for both SO₄ and BC, although BC aerosols generate significantly more warming due to their greater absorptivity (Extended Data Fig. 7a). The higher and thinner the atmosphere, the greater the

warming. It is important to note that higher temperatures may accelerate the catalytic reaction rates of ozone-depleting processes (Solomon et al., 1996, 1998), potentially resulting in greater ozone loss, although these effects are not considered in our simulations, which prescribe ozone concentrations. Aerosols in the lower stratosphere induce more warming around tropopause and upper troposphere, weakening the “freeze-drying” process and consequently increasing the stratospheric water vapor mixing ratio (Extended Data Fig. 7b; Mote et al., 1996; Randel et al., 2006). This water vapor increase contributes additional radiative forcing (Forster & Shine, 2002) and, more importantly, may catalyze ozone depletion through the formation of hydroxyl (OH) radicals (Solomon et al., 1986). The warming also leads to a larger increase in the saturation vapor pressure. Without significantly enhanced water vapor transport into the area, it is reasonable to expect a decrease in the relative humidity (Extended Data Fig. 7c). Similarly, cloud fraction decreases within the upper troposphere (Extended Data Fig. 7d). The lower we prescribe the BC aerosols, the more the cloud fraction decreases, likely due to the stability Iris hypothesis (Bony et al., 2016), which states that the increased stability reduces the magnitude of the radiatively driven clear-sky mass convergence at the height of anvil clouds, thus weakening convective detrainment at that height, leading to a reduction of the anvil coverage.

Beyond their direct radiative impacts, BC and SO₄ aerosols also influence stratospheric dynamics (e.g., Bednarz et al. 2023a, 2023b), warranting further consideration. Our preliminary results suggest that BC-induced warming in the upper stratosphere can modify both the Brewer-Dobson circulation (BDC) and the polar vortex. Unlike sulfate aerosols, which accelerate the BDC by enhancing tropical upwelling, BC-induced heating appears to slow it down, potentially altering stratospheric ozone transport and distribution (Extended Data Fig. 8). Seasonal variations in BC radiative heating may also strengthen the equator-to-pole temperature gradient in winter, reinforcing the polar vortex. This has implications for stratosphere-troposphere coupling, as a stronger polar vortex has been linked to a positive phase of the North Atlantic Oscillation, which in turn influences storm tracks, temperature anomalies, and precipitation patterns across midlatitude regions, particularly over Europe and North America (Baldwin et al., 2003; Hurrell, 1995). However, our model lacks interactive stratospheric chemistry and has a relatively low model top, limiting its ability to capture the full extent of these interactions, particularly given that the available monthly wind field outputs adopted to calculate residual circulation (Andrews et al., 1987; Salby, 1996) smooth out short-term fluctuations and lose fine-scale temporal details. Nonetheless, our findings suggest that BC-induced upper stratospheric warming may influence large-scale circulation patterns in ways distinct from SO₄ aerosol geoengineering.”

Lastly, I would suggest the authors acknowledge more recent studies in fully coupled models using BC, such as Haywood et al. (2022)

Haywood, J. M., Jones, A., Johnson, B. T., and McFarlane Smith, W.: Assessing the consequences of including aerosol absorption in potential stratospheric aerosol injection climate intervention strategies, *Atmos. Chem. Phys.*, 22, 6135–6150, <https://doi.org/10.5194/acp-22-6135-2022>, 2022.

Response: Thank you for the suggestion. We have incorporated references to more recent studies using fully coupled models, including Haywood et al. (2022), in the revised manuscript.

Reference

- Andrews, D. G., Holton, J. R., & Leovy, C. B. (1987). *Middle atmosphere dynamics*. Academic Press.
- Baldwin, M. P., Stephenson, D. B., Thompson, D. W., Dunkerton, T. J., Charlton, A. J., & O'Neill, A. (2003). Stratospheric memory and skill of extended-range weather forecasts. *Science*, **301**(5633), 636–640.
- Boucher, O., Randall, D., Artaxo, P., Bretherton, C., Feingold, G., Forster, P., et al. (2013). Clouds and aerosols. In T. F. Stocker, D. Qin, G.-K. Plattner, M. Tignor, S. K. Allen, J. Doschung, et al. (Eds.), *Climate change 2013: The physical science basis. Contribution of working group I to the fifth assessment report of the intergovernmental panel on climate change* (pp. 571–657). Cambridge University Press.
- Forster, P. M., & Shine, K. P. (2002). Assessing the climate impact of trends in stratospheric water vapor. *Geophysical Research Letters*, **29**(6), 10–1104.
- Forster, P. M., Richardson, T., Maycock, A. C., Smith, C. J., Samset, B. H., Myhre, G., & Schulz, M. (2016). Recommendations for diagnosing effective radiative forcing from climate models for CMIP6. *Journal of Geophysical Research: Atmospheres*, **121**, 12,460–12,475.
- Forster, P., Storelvmo, T., Armour, K., Collins, W., Dufresne, J., Frame, D., et al. (2021). The Earth's energy budget, climate feedbacks, and climate sensitivity. In V. Masson-Delmotte, P. Zhai, A. Pirani, S. L. Connors, C. Péan, S. Berger, et al. (Eds.), *Climate change 2021: The Physical Science Basis. Contribution of Working Group I to the Sixth Assessment Report of the Intergovernmental Panel on climate change*. Cambridge University Press.
- Gao, R.-S., Rosenlof, K. H., Kärcher, B., Tilmes, S., Toon, O. B., Maloney, C., and Yu, P. (2021). Toward practical stratospheric aerosol albedo modification: Solar-powered lofting. *Science Advances*, **7**, eabe3416.
- Haywood, J. M., Roberts, D. L., Slingo, A., Edwards, J. M., & Shine, K. P. (1997). General Circulation Model calculations of the direct radiative forcing by anthropogenic sulfate and fossil-fuel soot aerosol. *Journal of Climate*, **10**(7), 1562–1577.
- Haywood, J. M., & Ramaswamy, V. (1998). Global sensitivity studies of the direct radiative forcing due to anthropogenic sulfate and black carbon aerosols. *Journal of Geophysical Research*, **103**(D6), 6043–6058.
- Haywood, J. M., Ramaswamy, V., & Soden, B. J. (1999). Tropospheric aerosol climate forcing in clear-sky satellite observations over the oceans. *Science*, **283**(5406), 1299–1303.
- Haywood, J. M., Jones, A., Johnson, B. T., & McFarlane Smith, W. (2022). Assessing the consequences of including aerosol absorption in potential stratospheric aerosol injection climate intervention strategies. *Atmospheric Chemistry and Physics*, **22**(9), 6135–6150.
- Hurrell, J. W. (1995). Decadal trends in the North Atlantic Oscillation: Regional temperatures and precipitation. *Science*, **269**(5224), 676–679.
- Kiehl, J. T., & Briegleb, B. P. (1993). The relative roles of sulfate aerosols and greenhouse gases in climate forcing. *Science*, **260**(5106), 311–314.
- Mote, P. W., Rosenlof, K. H., McIntyre, M. E., Carr, E. S., Gille, J. C., Holton, J. R., et al. (1996). An atmospheric tape recorder: The imprint of tropical tropopause temperatures on stratospheric water vapor. *Journal of Geophysical Research*, **101**(D2), 3989–4006.

- Myhre, G., Shindell, D., Bréon, F.-M., Collins, W., Fuglestedt, J., Huang, J., et al. (2013). Anthropogenic and natural radiative forcing. In T. F. Stocker, D. Qin, G.-K. Plattner, M. Tignor, S. K. Allen, J. Doschung, et al. (Eds.), *Climate change 2013: The physical science basis. Contribution of working group I to the fifth assessment report of the intergovernmental panel on climate change* (pp. 659–740). Cambridge University Press.
- Randel, W. J., Wu, F., Vömel, H., Nedoluha, G. E., & Forster, P. (2006). Decreases in stratospheric water vapor after 2001: Links to changes in the tropical tropopause and the Brewer-Dobson circulation. *Journal of Geophysical Research*, **111**(D12), D12312.
- Salby, M. L. (1996). *Fundamentals of atmospheric physics*. Academic Press.
- Smith, C. J., Kramer, R. J., Myhre, G., Forster, P. M., Soden, B. J., Andrews, T., et al. (2018). Understanding rapid adjustments to diverse forcing agents. *Geophysical Research Letters*, **45**, 12,023–12,031.
- Smith, C. J., Kramer, R. J., Myhre, G., Alterskjær, K., Collins, W., Sima, A., et al. (2020). Effective radiative forcing and adjustments in CMIP6 models. *Atmospheric Chemistry and Physics*, **20**(16), 9591–9618.
- Solomon, S., Garcia, R. R., Rowland, F. S., & Wuebbles, D. J. (1986). On the depletion of Antarctic ozone. *Nature*, **321**(6072), 755-758.
- Solomon, S., Portmann, R. W., Garcia, R. R., Thomason, L. W., Poole, L. R., & McCormick, M. P. (1996). The role of aerosol variations in anthropogenic ozone depletion at northern midlatitudes. *Journal of Geophysical Research*, **101**(D3), 6713–6727.
- Solomon, S., Portmann, R. W., Garcia, R. R., Randel, W., Wu, F., Nagatani, R., et al. (1998). Ozone depletion at mid-latitudes: Coupling of volcanic aerosols and temperature variability to anthropogenic chlorine. *Geophysical Research Letters*, **25**(11), 1871–1874.

We thank the three anonymous reviewers for their insightful suggestions and constructive comments, which have significantly improved the quality of the manuscript. We are especially grateful for their engagement throughout the review process and are pleased that our revisions and responses were found helpful.

REVIEWERS' COMMENTS:

Reviewer #1 (Remarks to the Author):

After initially reading and reviewing the manuscript "Weakening the CO₂ Greenhouse Effect via Stratospheric Aerosol Injection" by He et al., I had some questions about the design of the study and the limitations of the modeling approach. Overall, I think my concerns were addressed quite well.

My first concern was with the title and the idea of "weakening the CO₂ greenhouse effect". I was initially unsure whether the proposed method of injecting absorbing aerosols into the upper atmosphere and the resulting radiative effects were directly related to the greenhouse effect of CO₂. The authors addressed this issue by repeating their analysis using offline radiative transfer calculations with CO₂ removed, demonstrating that CO₂ does indeed play a major role in the results. This was a valuable addition to the manuscript.

I also had questions about the feasibility of aerosol injection at higher altitudes, as is typically proposed, and the limitations inherent in the aerosol modeling approach. The authors have now acknowledged these limitations in the revised manuscript. They have also responded to my questions in detail, supported their answers with figures, and clarified several points in the text.

Overall, despite some limitations in the modeling approach (which the authors have appropriately acknowledged), I believe the manuscript is original and novel, presents interesting results, and is well written and thorough. I recommend it for publication.

Response: We sincerely thank the reviewer for their thoughtful and constructive comments, as well as for taking the time to review our manuscript. We are especially grateful that the reviewer found our responses helpful and that the revised manuscript addresses the initial concerns regarding the study design and modeling limitations.

Reviewer #2 (Remarks to the Author):

I appreciate the authors' substantial efforts to address my comments in my initial review, particularly regarding the technical and scientific limitations of this study. The manuscript now includes a more detailed description and discussion of uncertainties and overlooked factors related to aerosol transport, microphysical evolution, sedimentation, chemical and dynamical perturbations in the stratosphere, and the technical challenges of high-altitude aerosol deployment.

However, I remain concerned that these important caveats are not sufficiently reflected in the framing of the manuscript's main conclusions. The Abstract, Introduction, and Conclusion continue to strongly emphasize the potential advantages of the proposed upper-stratospheric absorbing aerosol method, including claims of "an order of magnitude greater efficiency" compared to traditional sulfate SAI, without clearly communicating that these results stem from highly idealized model experiments that do not include interactive aerosol sedimentation, dynamics, chemistry, or real-world operational considerations.

In my view, the current presentation could lead readers to overinterpret the practical viability and comparative advantage of the method. To prevent this, I recommend that the authors revise the manuscript to more explicitly qualify their main claims throughout, especially in the Abstract and Conclusion, making clear that the reported efficiency gains are specific to an "idealized model setup" and are subject to substantial scientific and technical uncertainties.

Provided that these revisions are made, i.e. clarifying the scope and limitations of the study in the main messaging (Abstract, Introduction, and Conclusion) to avoid overinterpretation of the analyses, I would be supportive of publication. I believe this work would then represent a valuable contribution to the ongoing exploration of SAI strategies and would help spark valuable discussion within the community.

Response: We thank the reviewer for their thoughtful and constructive feedback. In response to their concern, we have revised the Abstract, Introduction, and Conclusion to more clearly communicate that our findings are based on highly idealized model experiments. We now explicitly qualify our main conclusions by highlighting the absence of interactive aerosol microphysics, chemistry, sedimentation, and real-world operational considerations. We have also softened language around the claimed efficiency of upper-stratospheric absorbing aerosols and emphasized the substantial uncertainties and limitations associated with this approach. These revisions are intended to prevent overinterpretation of the practical viability of our proposed method and to ensure that the manuscript accurately reflects the scope of the study. We appreciate the reviewer's support for publication following these clarifications.

Reviewer #3 (Remarks to the Author):

I appreciate the detailed and thoughtful responses from the authors. I think they did an excellent job expanding the manuscript from its previous form to better highlight the limitations of their study.

However, I need to make one last note based on the expanded explanations: the size distributions for the bulk aerosol parametrizations that they use are incredibly small. 0.05 μm for sulfate and 0.01 μm for BC are very, very low values that would give rather different results in terms of TOA forcing compared to more "realistic" size distributions.

Mind you, this is not just pickiness: BC in nature is usually a much larger size (for instance, the size distribution from Australian fires was bimodal with 0.1 micron mean radius for the fine mode (one order larger than the ones used here) and most of the mass in a second mode with mean radius of 5 μm (see Yang et al., 2021). For sulfate, most of the mass would also be in accumulation/coarse mode (see Laakso et al., 2022). Longwave absorption is heavily dependant on this (see Dykema et al., 2016), therefore it is unlikely that the authors' results could ever be replicated in other models with any kind of interactive aerosol parametrizations.

Yang, X., Zhao, C., Yang, Y., Yan, X., and Fan, H.: Statistical aerosol properties associated with fire events from 2002 to 2019 and a case analysis in 2019 over Australia, *Atmos. Chem. Phys.*, 21, 3833–3853, <https://doi.org/10.5194/acp-21-3833-2021>, 2021.

Laakso, A., Niemeier, U., Visioni, D., Tilmes, S., and Kokkola, H.: Dependency of the impacts of geoengineering on the stratospheric sulfur injection strategy – Part 1: Intercomparison of modal and sectional aerosol modules, *Atmos. Chem. Phys.*, 22, 93–118, <https://doi.org/10.5194/acp-22-93-2022>, 2022.

Dykema, J. A., D. W. Keith, and F. N. Keutsch (2016), Improved aerosol radiative properties as a foundation for solar geoengineering risk assessment, *Geophys. Res. Lett.*, 43, 7758–7766, doi:10.1002/2016GL069258.

Response: We thank the reviewer for their thoughtful comments and for recognizing the improvements made in the revised manuscript. We appreciate the opportunity to clarify the aerosol size assumptions used in our study.

The aerosol size distributions in our simulations follow a lognormal distribution, with geometric mean radius and standard deviation values derived from established literature (Haywood & Ramaswamy, 1998). We acknowledge that the geometric mean radius of 0.05 μm for sulfate (SO_4) and 0.0118 μm for black carbon (BC) may appear small compared to some observational estimates. However, these values are not directly comparable to the modal or volume-based size distributions reported in studies such as Yang et al. (2021), which describe bimodal distributions including both fine and coarse modes.

To aid interpretation, we have computed the corresponding effective radius and volume distribution peak for the prescribed lognormal distributions ($\sigma = 2.0$):

Black carbon (BC):

- Geometric mean radius: 0.0118 μm
- Effective radius: $\sim 0.039 \mu\text{m}$
- Volume distribution peak: $\sim 0.050 \mu\text{m}$

Sulfate (SO₄):

- Geometric mean radius: 0.05 μm
- Effective radius: ~0.166 μm
- Volume distribution peak: ~0.211 μm

Thus, while the geometric mean radius appear small, the radiatively active portion of the distribution is significantly larger than might be inferred from the geometric mean alone. We fully agree with the reviewer that BC from biomass burning—such as the Australian fires analyzed in Yang et al. (2021)—can exhibit much larger sizes and bimodal distributions. However, the BC size prescribed in our simulations is more representative of fossil fuel combustion, which generally produces smaller particles and is more relevant to global-mean radiative forcing scenarios. For sulfate, the derived effective radius places the particles squarely within the accumulation mode and near the optimal size range (~0.2–0.3 μm) for shortwave scattering. Moreover, these same size assumptions are employed in the most recent GFDL model configurations (e.g., ESM4 and CM4; citation), as used in studies such as Gao et al. (2023) and Govardhan et al. (2023), supporting consistency with widely used climate modeling frameworks. We have now revised the aerosol size description in the manuscript to help avoid potential confusion regarding the size assumptions and their implications.

Regarding longwave absorption, we appreciate the reviewer’s point. As discussed by Dykema et al. (2016), larger sulfate particles can enhance longwave absorption, potentially influencing stratospheric heating, water vapor concentrations, and net radiative forcing. However, larger particles also tend to be less efficient shortwave scatterers, leading to complex tradeoffs in their radiative effects. While our simulations do not explicitly resolve these microphysical processes, we acknowledge that the fixed-size assumptions omit aerosol growth, coagulation, and sedimentation. As a result, our estimates of sulfate-driven longwave heating may be conservative and do not fully capture the dynamical feedbacks present in models with interactive aerosol schemes. Nonetheless, the use of idealized size distributions allows for controlled sensitivity experiments and is consistent with configurations widely adopted in prior studies.

Reference

Dykema, J. A., Keith, D. W. & Keutsch, F. N. Improved aerosol radiative properties as a foundation for solar geoengineering risk assessment. *Geophys. Res. Lett.* **43**, 7758–7766 (2016).

Gao, C. Y. et al. Volcanic drivers of stratospheric sulfur in GFDL ESM4. *J. Adv. Model Earth Syst.* **15**, e2022MS003532 (2023).

Govardhan, G., Paynter, D. & Ramaswamy, V. Effective radiative forcing of the internally mixed sulfate and black carbon aerosol in the GFDL AM4 model: The role played by other aerosol species. *J. Geophys. Res. Atmos.* **128**, e2023JD038481 (2023).

Haywood, J. M. & Ramaswamy, V. Global sensitivity studies of the direct radiative forcing due to anthropogenic sulfate and black carbon aerosols. *J. Geophys. Res.* **103**, 6043–6058 (1998).

Yang, X. et al. Statistical aerosol properties associated with fire events from 2002 to 2019 and a case analysis in 2019 over Australia. *Atmos. Chem. Phys.* **21**, 3833–3853 (2021).